# Fast Large Language Model Collaborative Decoding via Speculation

**Jiale Fu** [* 1 2]  **Yuchu Jiang** [* 1 2]  **Junkai Chen** [1 2]  **Jiaming Fan** [1 2]  **Xin Geng** [1 2]  **Xu Yang** [1 2]

## Abstract

Large Language Model (LLM) collaborative decoding techniques improve output quality by combining the outputs of multiple models at each generation step, but they incur high computational costs. In this paper, we introduce **Collaborative decoding via Speculation (CoS)**, a novel framework that accelerates collaborative decoding without compromising performance. Inspired by Speculative Decoding—where a small proposal model generates tokens sequentially, and a larger target model verifies them in parallel, our approach builds on two key insights: (1) the verification distribution can be the combined distribution of both the proposal and target models, and (2) alternating each model as the proposer and verifier can further enhance efficiency. We generalize this method to collaboration among $n$ models and theoretically prove that CoS is never slower than standard collaborative decoding, typically achieving faster speed. Extensive experiments demonstrate CoS is **1.11x–2.23x** faster than standard collaborative decoding without compromising generation quality. Our code is available at https://github.com/Kamichanw/CoS/.

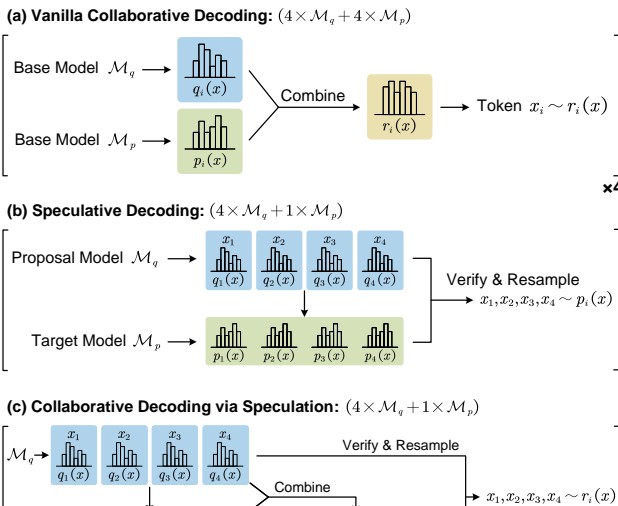

*Figure 1.* Comparison of (a) vanilla collaborative decoding, (b) speculative decoding, and (c) collaborative decoding via speculation. In (b) and (c), each discrete blue block represents a probability calculated by one forward pass of $\mathcal{M}_q$, while the continuous green block indicates the joint distribution requires only one forward pass of $\mathcal{M}_p$.

## 1. Introduction

Recently, large language models (LLMs) have demonstrated impressive performance across a wide range of tasks. Beyond advances in individual models—such as architectural innovations and training techniques—there is increasing interest in collaborative approaches involving multiple LLMs (Lu et al., 2024; Chen et al., 2025). A key class of these methods combines information from multiple models (e.g., probability distributions or logits) during token generation

to improve next-token prediction and capitalize on the complementary strengths of different models. For instance, ensembling methods (Yu et al., 2024; Huang et al., 2024; Yao et al., 2024) average prediction distributions from multiple models; contrastive decoding (Li et al., 2023; O'Brien & Lewis, 2023) improves generation quality and reduces hallucinations by subtracting the outputs of a smaller model from a larger one; and decoding-time realignment (Liu et al., 2024; Shi et al., 2024) jointly uses an aligned and an unaligned model during decoding to enable flexible control over alignment. In this paper, we refer to such approaches collectively as *collaborative decoding*.

Despite the significant progress in collaborative decoding, a key challenge persists: combining outputs from multiple models requires each model to perform a separate forward pass, which substantially slows down inference compared to using a single model. This raises a crucial question: can we speed up collaborative decoding without compromising quality? To address this, we propose **Collaborative**

[*]Equal contribution  [1]Southeast University  [2]Key Laboratory of New Generation Artificial Intelligence Technology and Its Interdisciplinary Applications (Southeast University), Ministry of Education, China. Correspondence to: Xu Yang <xuyang_palm@seu.edu.cn>.

*Proceedings of the 42nd International Conference on Machine Learning*, Vancouver, Canada. PMLR 267, 2025. Copyright 2025 by the author(s).

**decoding via Speculation (CoS)**, a novel framework that accelerates any form of collaborative decoding while maintaining output quality. The core idea of CoS comes from Speculative Decoding (SD) (Xia et al., 2023; Leviathan et al., 2023).

Speculative Decoding is a technique designed to accelerate LLM inference without sacrificing performance. As depicted in Figure 1 (b), it uses a smaller but more efficient proposal model $\mathcal{M}_q$ to rapidly generate proposal tokens, which are then verified in parallel by a larger target model $\mathcal{M}_p$. The target model accepts a subset of these proposal tokens, enabling the generation of multiple tokens in a single forward pass, thus significantly accelerating inference. Moreover, by employing specific acceptance-rejection criteria, the generated tokens can be considered as samples drawn from the distribution of the target model, thus ensuring the generation quality. In this paper, we extend speculative decoding to LLM collaborative decoding based on the following two observations.

**First, SD allows not only sampling from the target model's distribution,** but also sampling from any combined distribution of the proposal model and target model. In vanilla SD, the target model's distribution is directly employed for token verification and resampling, ensuring that the generated tokens align with the target model's distribution. Similarly, we find that if the combined distribution is used for verification and resampling, as illustrated in Figure 1 (c), the generated tokens will follow the combined distribution. We refer to this generalization of SD to collaborative decoding as *Naive-CoS*. Naive-CoS significantly reduces the number of model invocations required. For instance, as shown in Figure 1 (a), generating four tokens with the vanilla collaborative decoding necessitates four invocations to both $\mathcal{M}_q$ and $\mathcal{M}_p$, while Naive-CoS, in the optimal case, requires only four invocations to $\mathcal{M}_q$ and a single invocation to $\mathcal{M}_p$.

**Secondly, alternating each model as proposer and verifier can further accelerate the collaboration process.** In standard SD, the proposer and verifier are fixed, with one model consistently serving as the proposer and the other as the verifier. However, we observe that in the collaborative decoding setting, this static assignment is suboptimal, as it fails to fully leverage the bonus token. In SD, when all tokens from the proposal model are accepted by the target model, the target model will naturally generate an additional token, referred to as the *bonus token*. However, since the bonus token is drawn from the target model's distribution rather than the combined distribution, it cannot be directly appended to the output of collaborative decoding. A naive solution might be to discard the bonus token or to re-query the proposal model and compute the combined distribution. Instead, we propose a more efficient approach: treating the

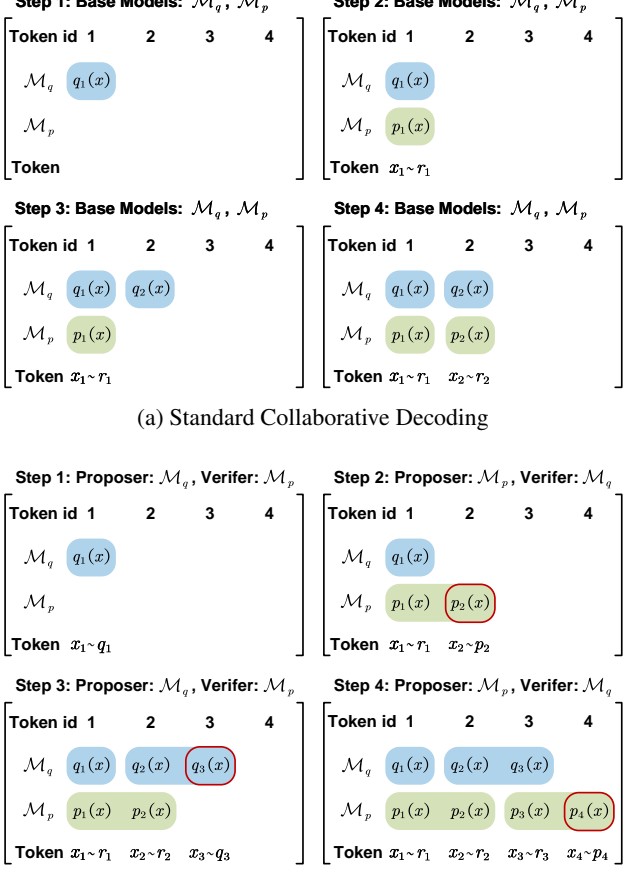

(a) Standard Collaborative Decoding

(b) Alternate Proposal Framework

*Figure 2.* The sketch of Alternate Proposal Framework. A continuous colored block indicates a single model invocation, with the bonus token highlighted in a red rounded box. Beginning from Step 2, $\mathcal{M}_q$ and $\mathcal{M}_p$ are invoked alternately. Each invocation involves both the verification of the current token and the generation of a bonus token. For clarity, we assume that the proposal length for each model is 1 and that all proposed tokens are accepted.

bonus token as a proposal from the target model, which is then verified by the proposer model. This insight leads to the *Alternate Proposal Framework*, illustrated in Figure 2. Combined with the Naive-CoS, the alternate proposal framework forms the proposed CoS. We further extend CoS to the general case of $n$-model collaboration in Section 3.4.

As shown in Figure 2(a), in standard collaborative decoding, each model invocation can generate only one probability distribution, so generating $n$ tokens requires $2n$ model invocations. With the alternate proposal framework (Figure Figure 2(b)), each invocation can generate two distributions in the optimal case, thereby doubling the generation efficiency.

We establish the effectiveness of CoS through both theoretical and experimental perspectives. Theoretically, we derive

an expected improvement factor to quantify its acceleration and prove that CoS is guaranteed to be at least as efficient as the standard collaborative decoding, typically achieving greater speed. Additionally, in the weighted ensemble setting—the most common collaborative decoding setting—we demonstrate that CoS maintains a provable lower bound on the acceptance rate, ensuring consistently high efficiency. Experimentally, we conduct extensive experiments across various tasks, including code generation, mathematical reasoning, multi-task understanding, and text summarization. Our evaluation covers multiple LLM pairs, including Llama, Vicuna, and Qwen series, under both two-model and three-model configurations. The results show that CoS consistently achieves the highest acceleration, with speedups of **1.34x–1.85x** for weighted ensemble and **1.11x–2.23x** for contrastive decoding.

In summary, our key contributions are as follows: (1) We extend speculative decoding to the collaborative decoding setting by refining its verification mechanism, introducing a Naive-CoS that significantly improves efficiency. (2) We incorporate an alternate proposal framework into the Naive-CoS to get the final CoS, further boosting inference speed. (3) Through extensive theoretical analysis and experimental evaluation, we demonstrate that our method achieves substantial acceleration while maintaining a lower bound, ensuring it never underperforms compared to standard collaborative decoding.

## 2. Related Work

**LLM Collaborative Decoding.** In this paper, LLM collaborative decoding refers to the integration of information from multiple LLMs when generating the next token. This typically involves combining the output probabilities or logits to compute a final probability distribution. This method can be used to improve overall performance, especially in solving complex tasks(Han et al., 2025; 2024). A common approach is model ensembling, which averages or applies weighted averaging to the probability distributions of multiple LLMs to derive the final sampling distribution (Yu et al., 2024; Huang et al., 2024; Yao et al., 2024). Studies have shown that this technique can enhance both performance and safety (Li et al., 2024a). Another method, contrastive decoding, is based on the observation that smaller models tend to produce noisier outputs. By subtracting the logits of a smaller model from those of a larger one, a cleaner and more reliable set of logits can be obtained, resulting in higher-quality outputs (Li et al., 2023; O'Brien & Lewis, 2023). Finally, decoding-time realignment enables flexible alignment with human preferences during decoding by linearly combining the logits of a human-aligned and an unaligned model, thereby balancing performance with alignment objectives (Liu et al., 2024; Shi et al., 2024).

Our proposed method, CoS, represents an orthogonal approach to existing collaborative decoding techniques. It is designed to substantially improve inference speed while preserving the benefits of collaboration. Importantly, CoS is not restricted to accelerating the three methods discussed above; its generality allows it to enhance the efficiency of any collaborative decoding approach, including those yet to be developed.

**Speculative Decoding.** Speculative decoding (Xia et al., 2023; Leviathan et al., 2023; Chen et al., 2023) can be categorized into two main areas: proposal model design and verification design. In the first category, proposal models are designed to generate tokens that are more likely to be accepted by the verifier. This includes independent proposal models, such as distillation-based method (Zhou et al., 2024) and target-informed models that incorporate information of verifier (Zhang et al., 2024; Elhoushi et al., 2024; Monea et al., 2023; Yi et al., 2024; Monea et al., 2023; Li et al., 2024b; Sun et al., 2024b).

The second category optimizes target model's verification process to improve decoding efficiency, following two main research directions. The first one increases proposal tokens and uses structured attention mechanisms (Miao et al., 2024; Cai et al., 2024; Li et al., 2024c; Gong et al., 2024) to validate multiple candidates simultaneously. The second direction modifies the verification strategy itself, employing methods like joint probability density estimation (Anonymous, 2025a), Monte Carlo tree search (Hu & Huang, 2024), and a linear binary classifier (Anonymous, 2025b).

The most closely related work is Speculative Contrastive Decoding (SCD) (Yuan et al., 2024), which combines outputs from both large and small models during the verification phase to form a contrastive decoding distribution. Operationally, SCD can be seen as a special case of Naive-CoS in the contrastive decoding setting. The method proposed in this paper differs from SCD in three key ways: (1) CoS is more broadly applicable and can accelerate any collaborative decoding approach; (2) it introduces an alternative proposal framework that further improves decoding speed; and (3) it provides a complete theoretical analysis, ensuring inference efficiency comparable to standard collaborative decoding methods.

## 3. Collaborative Decoding via Speculation

### 3.1. Speculative Decoding

Unlike other acceleration methods (Sun et al., 2024a), speculative decoding (SD) is a technique designed to speed up inference while maintaining the quality of generated outputs. It involves two phases: the proposal phase and the verification phase. During the proposal phase, a lightweight proposal model sequentially generates proposal tokens. In

the verification phase, a larger target model verifies these tokens in parallel. Furthermore, by incorporating appropriate acceptance-rejection criteria, the technique ensures that the generated tokens align precisely with the target model's distribution, thus maintaining high-quality results.

Specifically, in the proposal phase, the proposal model $\mathcal{M}_q$ generates a sequence of length $\gamma$, denoted as:

$$(x_{i+1}, x_{i+2}, \ldots, x_{i+\gamma}) \sim \prod_{j=1}^{\gamma} q_{i+j}(x). \qquad (1)$$

Here, $x_{i+j}$ represents the token generated at position $i + j$, and $q_{i+j}(x) \triangleq q(x_{i+j} \mid x_{\leq i+j-1})$ is the conditional probability distribution computed by $\mathcal{M}_q$ over $x_{i+j}$, given the previously generated sequence $x_{\leq i+j-1}$.

In the verification phase, the target model $\mathcal{M}_p$ executes a forward pass, producing $\gamma + 1$ target distributions: $p_{i+1}(x), \ldots, p_{i+\gamma}(x), p_{i+\gamma+1}(x)$. The first $\gamma$ distributions are subsequently used to validate the proposal tokens generated in the proposal phase. Specifically, a proposal token $x_{i+j}$ is accepted if the following condition holds:

$$u_j \leq \min\left(1, \frac{p_{i+j}(x)}{q_{i+j}(x)}\right) \qquad (2)$$

where $u_j \sim U(0,1)$ represents a uniformly distributed random variable. If the token $x_{i+j}$ is rejected, the subsequent tokens $x_{i+j+1}, \ldots, x_{i+\gamma+1}$ are discarded, and $x_{i+j}$ is sampled from the distribution $\mathrm{norm}(\max(0, p_{i+j} - q_{i+j}))$. If all $\gamma$ tokens are accepted, an additional token is directly sampled from $p_{i+\gamma+1}$ and appended to the generated sequence, referred to as the *bonus token* in our paper.

By iteratively alternating between the proposal and verification phases, SD improves inference speed while ensuring the generated tokens align with the target model's distribution.

### 3.2. Naive-CoS

As discussed above, vanilla SD can only accelerate the inference of a single model. In this subsection, we will introduce how to apply SD to scenarios involving an arbitrary combination of two models. Specifically, let $q_i(x)$ and $p_i(x)$ denote the distributions of token $x_i$ given by the proposal model and the target model, respectively, and let $l_i^q$ and $l_i^p$ be the corresponding logits. Then, the combined distribution $r_i(x)$ can be expressed as

$$r_i(x) = \mathcal{C}(q_i(x), p_i(x)) \text{ or } \mathcal{C}'(l_i^q, l_i^p), \qquad (3)$$

where $\mathcal{C}(\cdot)$ represents the combination function at the probability level, while $\mathcal{C}'(\cdot)$ is at logits level. For example, the common weighted ensemble that uses probability for weighted summation can be expressed as

$$r_i(x) = \mathcal{C}(q_i(x), p_i(x)) = \lambda q_i(x) + (1 - \lambda) p_i(x), \quad (4)$$

while contrastive decoding can be represented as

$$r_i(x) = \mathcal{C}'(l_i^q, l_i^p) = \mathrm{Softmax}(l_i^p - \mu l_i^q). \qquad (5)$$

We note that by making slight modifications to the vanilla SD, the generated tokens can align with the combined distribution. Specifically, before verification, we first compute the combined distribution $r_i(x)$ and update the verification formula in Equation (2) as follows:

$$u_j \leq \min\left(1, \frac{r_{i+j}(x)}{q_{i+j}(x)}\right). \qquad (6)$$

Then, if the token is rejected, we resample $x_{i+j}$ from the distribution $\mathrm{norm}(\max(0, r_{i+j} - q_{i+j}))$.

We theoretically prove the correctness of Naive-CoS, that is, the tokens generated by the above sampling process precisely align with the combined distribution, with an acceptance rate $\alpha$ of

$$\alpha = 1 - \frac{1}{2} D_{\mathrm{TV}}(q, r), \qquad (7)$$

where $D_{\mathrm{TV}}(q, r)$ is the total variation distance, defined as $D_{\mathrm{TV}}(q, r) = \sum_{x \in \mathcal{V}} |q(x) - r(x)|$, where $\mathcal{V}$ is the set of all tokens. The proof is provided in Appendix A.1.

**Analysis of speed improvement.** In speculative decoding, inference speed is predominantly influenced by the acceptance rate $\alpha$, with a higher acceptance rate leading to more substantial speed improvements. In this part, we first analyze the theoretical speed improvement when $\alpha$ is known. When $\alpha$ is unknown, we focus on weighted ensemble scenario and provide a lower bound for $\alpha$. With this bound, we derive a series of favorable acceleration properties.

**Theorem 3.1.** *Let $\gamma$ be the proposal length and $c$ be the cost coefficient, defined as the ratio between the time for a single invocation of the proposal model and the target model. Then, the expected speed improvement factor is $\frac{(1-\alpha^\gamma)(1+c)}{(1-\alpha)(1+c\gamma)}$.*

The proof of Theorem 3.1 is in Appendix A.2. Theorem 3.1 provides the speed improvement factor when $\alpha$ is known. However, in most cases, $\alpha$ is unknown and requires extensive experiments to estimate. Nevertheless, we find that in the weighted ensemble scenario, which is the most common collaborative decoding, $\alpha$ has a lower bound.

**Theorem 3.2.** *If $\mathcal{C}(p, q) = \lambda q(x) + (1 - \lambda)p(x)$ and $q(x)$ is the proposal model. Then $\alpha$ has a lower bound of $\lambda$.*

*Proof.* We have $\alpha = \sum_{x \in \mathcal{V}} q(x) \min\left(1, \frac{r(x)}{q(x)}\right)$, then $\alpha = \sum_{x \in \mathcal{V}} q(x) \min\left(1, \lambda + (1 - \lambda)\frac{p(x)}{q(x)}\right)$, and then we get $\alpha \geq \sum_{x \in \mathcal{V}} \lambda q(x) = \lambda$.

The equality holds if and only if $p(x)q(x) = 0$ for all $x \in \mathcal{V}$, which means that $p(x)$ and $q(x)$ do not overlap. $\square$

**Corollary 3.3.** *Assume that $\mathcal{C}(p, q) = \lambda q(x) + (1 - \lambda)p(x)$, and that $\mathcal{M}_q$ and $\mathcal{M}_p$ have comparable parameters. Then, by selecting an appropriate proposal model, $\alpha$ has a lower bound of at least 0.5.*

*Proof.* Since $\mathcal{M}_q$ and $\mathcal{M}_p$ have comparable parameters, either can serve as the proposal model. Therefore, $\alpha$ has a lower bound of $\max(\lambda, 1 - \lambda)$, which is at least 0.5. □

By utilizing the lower bound property, we demonstrate that the proposed Naive-CoS is guaranteed to be no slower than the weighted ensemble approach, and it is typically faster.

**Corollary 3.4.** *Assume that $\mathcal{C}(p, q) = \lambda q(x) + (1 - \lambda)p(x)$, then if $\lambda > \frac{c}{1+c}$, there exists a value of $\gamma$ that enhances the inference speed.*

*Proof.* Consider $\gamma = 2$, and solve the inequality $\frac{(1-\alpha^\gamma)(1+c)}{(1-\alpha)(1+c\gamma)} > 1$. Solving this inequality yields $\alpha > \frac{c}{1+c}$. Since $\alpha \geq \lambda$ follows, establishing the corollary. □

**Corollary 3.5.** *Assume that $\mathcal{C}(p, q) = \lambda q(x) + (1 - \lambda)p(x)$. Then, for any $\lambda$, there exists a value of $\gamma$ such that the speed of the Naive-CoS is not slower than the vanilla weighted ensemble, and it is almost always faster.*

*Proof.* As stated in Corollary 3.4, if $\lambda > \frac{c}{1+c}$, there exists a value of $\gamma$ that enhances the inference speed. If we swap the proposer and the verifier, the condition for acceleration changes to $1 - \lambda > \frac{1/c}{1+1/c}$, which simplifies to $\lambda < \frac{c}{1+c}$.

If $\lambda \neq \frac{c}{1+c}$, either $\lambda > \frac{c}{1+c}$ or $\lambda < \frac{c}{1+c}$ must hold, which ensures a speedup. Otherwise, if $\lambda = \frac{c}{1+c}$, then $\alpha \geq \lambda = \frac{(1-\alpha^\gamma)(1+c)}{(1-\alpha)(1+c\gamma)}$, which ensures that the speed does not decline. Moreover, equality only holds when the two distributions do not overlap, which is almost impossible in practice. □

**Interpretable quality-speed tradeoff in vanilla SD.** In SD, some studies focus on relaxing the acceptance criteria for a higher acceptance rate to achieve faster inference, such as lossy SD (Zhou et al., 2024) and typical acceptance (Cai et al., 2024). However, these methods often lack interpretability, that is, we do not know which distribution the generated tokens will follow. We find that Naive-CoS can naturally be an interpretable strategy for adjusting the quality-speed tradeoff in vanilla SD.

Specifically, we apply a weighted ensemble using the proposal and target models in SD. In this setup, when $\lambda = 0$, the combined distribution aligns exactly with the target distribution, reducing the method to standard SD. For $\lambda > 0$, the acceptance rate is guaranteed to have a lower bound and greater than that of vanilla SD, as demonstrated in the proof of Theorem 3.2, leading to greater acceleration. However, incorporating less precise information from a smaller model

can introduce some performance degradation. This trade-off provides a mechanism to balance quality and speed in speculative decoding.

In contrast to existing approaches, this method improves interpretability. This is because we know the distribution of the generated tokens after relaxation, i.e. combined distribution defined in Equation (4). This allows us to design proposal models that accelerate inference without compromising performance, and potentially even enhance the model's capabilities in specific areas. For instance, some research suggests that ensembling a smaller model appropriately can improve safety (Wang et al., 2024; Li et al., 2024a). The experimental results are shown in Appendix C.1.

### 3.3. Alternate Proposal Framework

In Section 3.2, we explore the application of speculative decoding to LLM collaboration. However, we don't consider the bonus token, that is, the additional token generated when all proposal tokens are accepted. This is because the bonus token follows the distribution of the verifier rather than the combined distribution and can not be directly appended to the output sequence. In this subsection, we introduce a collaboration framework, termed the *alternate proposal framework*, which effectively leverages the bonus token and demonstrates superior performance.

As shown in Figure 2, in the alternate proposal framework, the generation of a bonus token is treated as a proposal from the current verifier, which is subsequently verified by the current proposer. Specifically, let the proposer be denoted as $\mathcal{M}_q$ and the verifier as $\mathcal{M}_p$ with proposal lengths $\gamma_q$ and $\gamma_p$, respectively. If all tokens proposed by $\mathcal{M}_q$ are accepted, a total of $\gamma_q + 1$ tokens will be generated. The first $\gamma_q$ tokens follows the distribution $r_{i+j}(x) = \mathcal{C}(q_{i+j}(x), p_{i+j}(x))$, for $j = 1, \ldots, \gamma_q$, while the $\gamma_q + 1$-th token is drawn from $p_{i+\gamma_q+1}(x)$. At this stage, the $\gamma_q + 1$-th token, referred to as bonus token, is treated as the initial token in $\mathcal{M}_p$'s proposal. Subsequently, $\mathcal{M}_p$ will generate an additional $\gamma_p - 1$ tokens to complete its proposal.

If any proposed tokens are rejected and no bonus token is generated, the default proposal model will take over as the proposal model. This default model is predefined and fixed. As outlined above, the two models alternate as proposers during the decoding process, which is why this approach is called the alternate proposal framework. The pseudocode for this framework is provided in Algorithm 1.

**Analysis of speed improvement.** We now analyze the speed improvement achieved by the alternative proposal framework. For the sake of clarity, we focus on a single cycle, which encompasses one proposal and one verification. In this cycle, both the proposer and verifier are fixed. Given that the decoding process is composed of multiple such

cycles, the overall decoding performance can be inferred from the behavior of a single cycle.

First, similar to Theorem 3.1, we provide the expected speed improvement factor for the alternate proposal framework.

**Theorem 3.6.** *Let $\mathcal{M}_q$ be the proposer and $\mathcal{M}_p$ be the verifier, then the expected speed improvement factor of the alternate proposal framework is $\frac{(1-\alpha^{\gamma_q})(1+c)}{(1-\alpha)(1+c\gamma_q-\alpha^{\gamma_p}c)}$.*

*Proof.* When the bonus token is generated, the proposer only needs to generate $\gamma_q - 1$ new tokens; otherwise, it must generate $\gamma_q$ tokens. The probability of generating the bonus token is the probability that all proposal tokens in the last cycle were accepted, which is $\alpha^{\gamma_p}$. Therefore, the expected time spent on proposal and verification is $\alpha^{\gamma_p}(1 + c(\gamma_q - 1)) + (1 - \alpha^{\gamma_p})(1 + c\gamma_q)$. Then the factor can be derived following the process in Appendix A.2. □

In Corollary 3.5, we proved that in the weighted ensemble scenario, the Naive-CoS is never slower than the vanilla collaborative decoding and is typically faster. In this subsection, with the alternate proposal framework, we extend this conclusion to any form of two-model collaboration.

**Corollary 3.7.** *For any two models, there exist values of $\gamma_q$ and $\gamma_p$ such that the speed of the alternate proposal framework is never slower than the vanilla collaboration and is almost always faster.*

*Proof.* Consider $\gamma_q = \gamma_p = 1$, then $\frac{(1-\alpha^{\gamma_q})(1+c)}{(1-\alpha)(1+c\gamma_q-\alpha^{\gamma_p}c)} \geq 1$ holds universally. The equality holds only when $c = 0$ or $\alpha = 0$. However, $c > 0$ because the execution time of the proposal model is non-negligible, and $\alpha > 0$ holds unless the proposal distribution and the combined distribution do not overlap, which is almost impossible. □

An intuitive interpretation of Corollary 3.7 is that when $\gamma_q = \gamma_p = 1$, even in the worst-case scenario—where all proposal tokens are rejected—each token generation still requires only one proposal and one verification. This results in the same number of model invocations as the standard collaborative decoding. In practice, however, it is rare for all proposal tokens to be rejected. Once a token is accepted, the collaboration process becomes more efficient.

### 3.4. Generalize to More Models

In this subsection, we extend CoS to the $n$-model collaboration scenario. The core principles remain similar to the two-model case, with acceleration driven by two key factors. First, each model can score the proposals of other models in parallel, where scoring refers to computing the probabil-

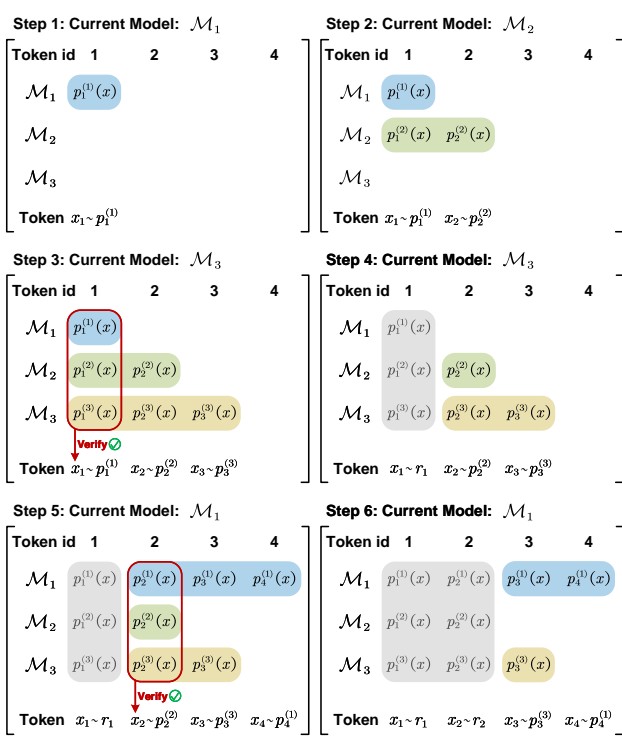

*Figure 3.* The sketch of CoS in three-model collaboration scenario. The colored boxes represent the stored probability distributions, while the gray boxes represent the discarded ones. Each invocation involves scoring the current proposal tokens and generating a bonus token. For clarity, we assume that the proposal length for each model is 1 and that all proposed tokens are accepted.

ity distribution of a proposal from other models.[1] Second, during scoring, a model can naturally generate a bonus token, which further improves efficiency. We illustrate the CoS process in the $n$-model scenario with a simple example, while detailed pseudocode and a general visualization are provided in Appendix B.2.

As shown in Figure 3, the process begins in step 1 with the default proposal model, $\mathcal{M}_1$, generating a proposal token $x_1$. In step 2, $\mathcal{M}_2$ scores $x_1$ while simultaneously generating a bonus token $x_2$. Similarly, in step 3, $\mathcal{M}_3$ scores both $x_1$ and $x_2$ in parallel and produces another bonus token, $x_3$. At this point, $x_1$ has been scored by both $\mathcal{M}_2$ and $\mathcal{M}_3$, enabling the computation of its combined distribution $r_1(x)$ for verification. The associated distributions $p_1^{(1)}(x)$, $p_2^{(1)}(x)$, $p_3^{(1)}(x)$ are no longer needed and are discarded.

If $x_1$ is accepted, $\mathcal{M}_1$ computes $p_2^{(1)}(x)$, $p_3^{(1)}(x)$, $p_4^{(1)}(x)$ in parallel as shown in step 5, allowing verification of $x_2$. Oth-

---

[1]We use the term "scoring" rather than "verification" because, unlike in the two-model case, scoring does not immediately trigger verification; instead, verification occurs only after all models have scored a token.

erwise, if $x_1$ is rejected, all stored distributions are cleared, and $\mathcal{M}_1$ generates a new proposal, similar to step 1.

# 4. Experiments

## 4.1. Experimental Setups

**Datasets and evaluation.** We test CoS across multiple tasks including code generation, mathematical reasoning, multi-task understanding, and text summarization on HumanEval (Chen et al., 2021), GSM8K (Cobbe et al., 2021), MMLU (Hendrycks et al., 2021), and CNNDM (See et al., 2017), respectively. We measure each method's speed by the average tokens generated per second and compute the speedup ratio relative to the standard collaborative decoding. All experiments are conducted on RTX 3090, except for evaluations involving the Llama-Vicuna model pair, which use the A6000 GPU. Additionally, we also test on the Ascend 910B3 NPU; the corresponding results are shown in Table 9 and Table 10.

**Combination functions and methods.** We experiment with two combination functions: weighted ensemble (WE) at the distribution level (Equation (4)) and contrastive decoding (CD) at the logits level (Equation (5)). For WE, in the two-model case, we set $\lambda = 0.5$ and temperature $T = 1$; in the three-model case, each model's coefficient was set to $1/3$. For CD, we set $\mu = 0.1$, which is the most common setting, and set $T$ to both 0 and 1. WE with $T = 0$ is not tested due to its uncommon use, as it leads to a one-hot distribution, reducing information. Among two combination functions, four methods are compared: (1) the standard collaborative decoding (**WE, CD**); (1) parallel collaborative decoding (**WE-P, CD-P**); (2) an accelerated version with speculative decoding (SD), using the smallest model as the proposal and the combined distribution as the target (**WE-SD, CD-SD**); and (3) CoS (**WE-CoS, CD-CoS**). Since SCD is equivalent to Naive-CoS, its results are included in our ablation on alternative proposal frameworks (Appendix C.4).

**Model pair configuration.** We experiment on different types of LLMs, including Llama-2 (Touvron et al., 2023; Miao et al., 2024), Vicuna (Zheng et al., 2023), Llama-3 (Dubey et al., 2024), Qwen-2.5 (Team, 2024), and OPT (Zhang et al., 2022). Model pair configurations for each combination function are in Table 1. We also test a three-model collaboration using Qwen2.5-1.5B-Instruct and its code and math versions in the WE setting.

**Configuration of $\gamma$.** The proposal length $\gamma$ is the only hyperparameter in SD, affecting the algorithm's acceleration. In the two-model CoS setting, $\gamma$ corresponds to the proposal length of the smaller model, with the larger model fixed at 1. For simplicity, we refer to the smaller model with $\gamma > 1$ as the proposal model of CoS, since it typically serves this role. We tested $\gamma = 5$ and $\gamma = 1$ for CoS and SD speeds,

*Table 1.* Model pair configuration. The first column represents the name of the corresponding model pair for simplicity.

| Name | $\mathcal{M}_q$ | $\mathcal{M}_p$ |
|---|---|---|
| *Weight Ensemble (WE)* | | |
| Llama-Vicuna | Llama-2-7B | Vicuna-7B-V1.5 |
| Qwen-3b | Qwen2.5-3B-Instruct | Qwen2.5-Coder-3B-Instruct |
| Qwen-1.5b | Qwen2.5-1.5B-Instruct | Qwen2.5-Coder-1.5B-Instruct |
| *Contrastive Decoding (CD)* | | |
| Llama-3 | Llama-3.2-1B | Llama-3.1-8B-Instruct |
| Llama-2 | Llama-68M | Llama-2-7B |
| OPT | OPT-125M | OPT-13B |

reporting the optimal results. $\gamma = 5$ is the common setting, while $\gamma = 1$ ensures acceleration (Corollary 3.7). In the three-model CoS, all models have a proposal length of 1.

## 4.2. Main Results

Table 2 and Table 3 display the speedup ratios for each method relative to the standard collaborative decoding in the WE and CD settings, respectively.[2] From these two tables, we have the following findings. First, CoS not only consistently achieves the highest speedup in all settings, it also gets speedup across all settings, which supports the findings in Corollary 3.7. In contrast, SD may reduce the collaboration speed in some cases. For example, when using the Llama-2 model pair with $T = 1$ on HumanEval in Table 3, applying SD reduces the speed to 0.94x of the standard collaboration. A similar speed reduction was also observed in the three-model scenario in Table 2. This is because vanilla SD does not inherently ensure acceleration. When the acceptance rate is low, SD may perform slower than standard decoding.

Second, compared to the CD scenario, the WE scenario ensures a higher minimum speedup for CoS. In the two-model case, CoS achieves a minimum speedup of 1.34x, while in the three-model case, it reaches at least 1.27x. In contrast, the CD scenario has a speedup as low as 1.11x. This difference arises because CoS maintains a consistently high acceptance rate in the WE scenario, as outlined in Corollary 3.3.

Third, the speedup varies across tasks and is influenced by the determinism of task outputs. For example, in the WE scenario, CoS achieved the highest speedup on HumanEval, averaging 1.65x, as code generation demands strictly formatted outputs. Conversely, CoS has a lower speedup of 1.36x on a text summarization task, where output flexibility is higher. This difference stems from the alignment between the proposal and target models: in highly deterministic tasks, their outputs exhibit greater similarity, leading to a higher

---

[2]The results of OPT model pair are shown in Appendix C.2.

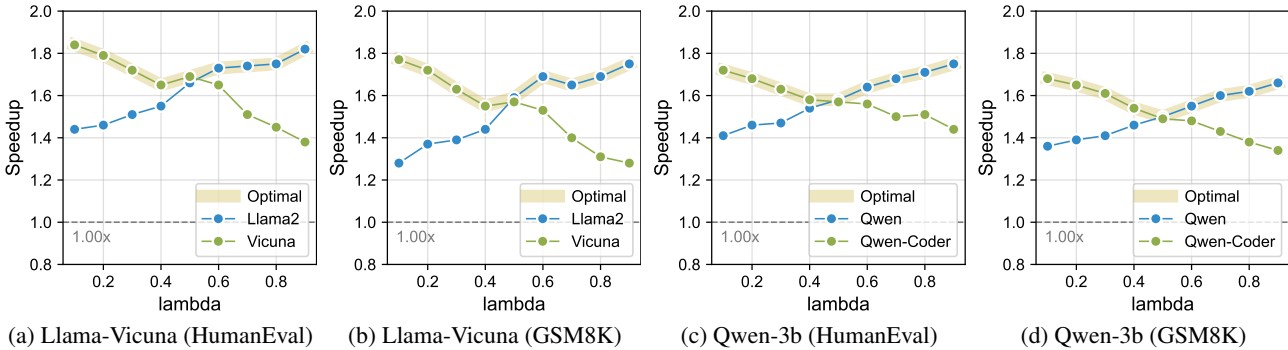

| (a) Llama-Vicuna (HumanEval) | (b) Llama-Vicuna (GSM8K) | (c) Qwen-3b (HumanEval) | (d) Qwen-3b (GSM8K) |

*Figure 4.* Comparison of speedup ratios for different $\lambda$ in WE across diverse settings. The blue and green lines represent the speedup ratios when the corresponding models serve as the proposal model, while the shaded region highlights the maximum speedup between the two.

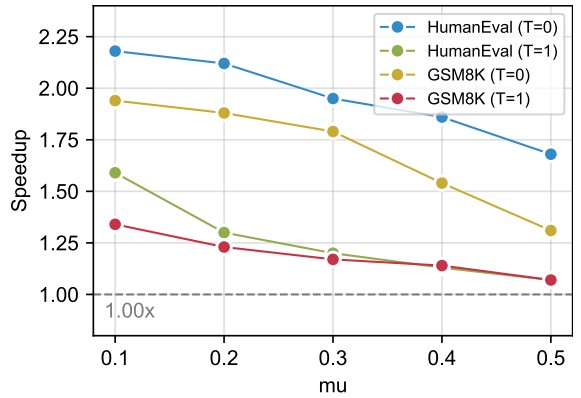

*Figure 5.* Comparison of speedup ratios for different $\mu$ in CD across different temperatures and datasets.

acceptance rate and, consequently, stronger acceleration.

### 4.3. Analysis

**Impact of proposal length $\gamma$.** We evaluate the influence of various $\gamma$ values, ranging from 1 to 5, on speedup across both the WE and CD scenarios, utilizing the HumanEval and GSM8K datasets. The results in Figure 6 show that when the models are similar in size, the speedup ratio remains stable across $\gamma$ values, as seen in the WE scenario Figure 6 (a). This is because the high cost of invoking the proposal model offsets the speedup from increasing $\gamma$. However, when the models differ significantly in size, the speedup ratio varies considerably with $\gamma$, as shown in the CD scenario Figure 6 (b) and (c).

Additionally, we observe that speedup initially increases with $\gamma$ before decreasing. For example, in the experiment with the Llama-3 model pair on GSM8K (Figure 6 (b)), speedup improves as $\gamma$ rises from 1 to 5, peaks at $\gamma = 5$, and then declines. This behavior is explained by two factors: increasing $\gamma$ boosts the expected number of accepted

*Table 2.* The speedup ratio of each method in WE setting. The method with the optimal speedup is highlighted in **bold**.

| | Method | HumanEval | GSM8K | MMLU | CNNDM |
|---|---|---|---|---|---|
| Llama Vicuna | WE | 1.00x | 1.00x | 1.00x | 1.00x |
| | WE-P | 0.69x | 0.73x | 0.70x | 0.75x |
| | SD | 1.27x | 1.21x | 1.19x | 1.15x |
| | CoS | **1.58x** | **1.52x** | **1.41x** | **1.46x** |
| Qwen-3b | WE | 1.00x | 1.00x | 1.00x | 1.00x |
| | WE-P | 0.74x | 0.79x | 0.79x | 0.77 |
| | SD | 1.13x | 1.06x | 1.09x | 1.08x |
| | CoS | **1.62x** | **1.52x** | **1.42x** | **1.38x** |
| Qwen-1.5b | WE | 1.00x | 1.00x | 1.00x | 1.00x |
| | WE-P | 0.63x | 0.62x | 0.64x | 0.63x |
| | SD | 1.11x | 1.13x | 1.08x | 1.10x |
| | CoS | **1.56x** | **1.46x** | **1.34x** | **1.35x** |
| Qwen-1.5b (3 Model) | WE | 1.00x | 1.00x | 1.00x | 1.00x |
| | WE-P | 0.54x | 0.73x | 0.80x | 0.82x |
| | SD | 0.96x | 0.92x | 0.98x | 0.95x |
| | CoS | **1.85x** | **1.53x** | **1.38x** | **1.27x** |

tokens, which improves acceleration; while later proposal tokens depend on earlier, unverified tokens, making them less accurate and more likely to be rejected, which wastes computation. Thus, the optimal speedup is achieved at a specific $\gamma$. In some cases, however, speedup either monotonically increases or decreases due to high or low acceptance rates. For instance, this is observed in the experiment experiments with the Llama-3 pair on HumanEval (Figure 6 (b)) and the Llama-2 pair on HumanEval (Figure 6 (c)).

**Speedup ratio for different weight $\lambda$ in WE.** We exam-

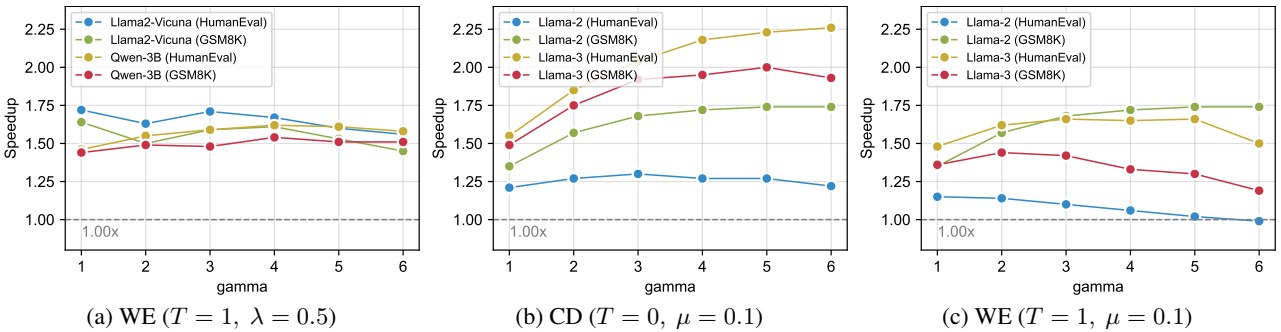

*Figure 6.* Comparison of speedup ratios for different $\gamma$ across different settings.

*Table 3.* The speedup ratio of each method in CD setting.

| | T | Method | HumanEval | GSM8K | MMLU | CNNDM |
|---|---|---|---|---|---|---|
| Llama-3 | 0 | CD | 1.00x | 1.00x | 1.00x | 1.00x |
| | | CD-P | 0.41x | 0.40x | 0.41x | 0.41x |
| | | SD | 2.04x | 1.81x | 1.52x | 1.58x |
| | | CoS | **2.23x** | **2.00x** | **1.77x** | **1.61x** |
| | 1 | CD | 1.00x | 1.00x | 1.00x | 1.00x |
| | | CD-P | 0.39x | 0.41x | 0.42x | 0.41x |
| | | SD | 1.55x | 1.21x | 1.20x | 1.07x |
| | | CoS | **1.65x** | **1.44x** | **1.31x** | **1.18x** |
| Llama-2 | 0 | CD | 1.00x | 1.00x | 1.00x | 1.00x |
| | | CD-P | 0.59x | 0.50x | 0.54x | 0.48x |
| | | SD | 1.15x | 1.62x | 1.08x | 0.93x |
| | | CoS | **1.26x** | **1.65x** | **1.68x** | **1.30x** |
| | 1 | CD | 1.00x | 1.00x | 1.00x | 1.00x |
| | | CD-P | 0.56x | 0.51x | 0.53x | 0.49x |
| | | SD | 0.94x | 1.16x | 1.23x | 1.10x |
| | | CoS | **1.15x** | **1.20x** | **1.37x** | **1.11x** |

ine the speedup effect of CoS when $\lambda$ takes values other than just 0.5. Specifically, we conduct experiments with $\lambda$ values ranging from 0.1 to 0.9, using the Llama-Vicuna and Qwen-3b model pairs on the HumanEval and GSM8K datasets. The results, presented in Figure 4, show that CoS consistently achieves a high speedup of at least 1.5x across all tested $\lambda$ values. This consistent speedup occurs because, when the two models are of similar sizes, for any $\lambda$, an appropriate proposal model can be selected to maintain a high acceptance rate during CoS process (as explained in Corollary 3.3), ensuring the observed speedup.

**Speedup ratio for different weight values of $\mu$ in CD.** Similarly, we examine the speedup effect of CD when $\mu$ takes other values. Specifically, we conduct experiments with $\mu$ ranging from 0.1 to 0.5, using the Llama-3 model pair on the HumanEval and GSM8K datasets. The results, presented in Figure 5, show that CoS consistently accelerates the CD

process across all tested $\mu$.

Furthermore, we find that an increase in $\mu$ results in a reduced speedup. This occurs because CD computes the combined distribution by subtracting the proposal model's information from the target model's. As $\mu$ grows, the gap between these distributions widens, lowering the acceptance rate (Equation (7)). Despite this, the speedup ratio remains above 1.00x, confirming that CD-CoS always accelerates, consistent with Corollary 3.7.

## 5. Conclusion

This paper introduces Collaborative Decoding via Speculation (CoS), an extension of speculative decoding that accelerates LLM collaborative decoding while maintaining output quality. CoS refines the verification mechanism for direct collaborative sampling and introduces an alternate proposal framework to further boost efficiency. We demonstrate the effectiveness of CoS through both theoretical analysis and empirical validation.

## Acknowledgements

This work is supported by the National Science Foundation of China (62206048), the Natural Science Foundation of Jiangsu Province (BK20220819), and the Fundamental Research Funds for the Central Universities (2242024k30035). Additional support was provided by the Big Data Computing Center of Southeast University and the SEU Kunpeng & Ascend center of cultivation.

## Impact Statement

This paper presents work whose goal is to advance the field of Deep Learning. There are many potential societal consequences of our work, none which we feel must be specifically highlighted here.

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

# A. Mathematical Proofs

## A.1. Correctness of CoS

Assume that in speculative decoding, the proposal distribution is $q(x)$, the target distribution is $p(x)$, and the combined distribution is $r(x) = \mathcal{C}(q(x), p(x))$. Referring to the proof of speculative decoding correctness (Leviathan et al., 2023), we now prove that the sampling method described in Section 3.2 ensures that the generated tokens align with the combined distribution $r(x)$.

First, we have:

$$P(x = x') = P(\text{proposal } x' \text{ accepted}) \cdot P(\text{proposal} = x') + P(\text{proposal rejected}) \cdot P(\text{resampled} = x') \tag{8}$$

By definition, $P(\text{proposal } x' \text{ accepted}) = \min\left(1, \frac{r(x')}{q(x')}\right)$ and $P(\text{proposal} = x') = q(x')$.

$$
\begin{aligned}
P(\text{proposal rejected}) &= \sum_{x \in \mathcal{V}} q(x) \left[1 - \min\left(1, \frac{r(x)}{q(x)}\right)\right] \\
&= \sum_{x \in \mathcal{V}} \left[q(x) - \min\left(q(x), r(x)\right)\right] \\
&= \sum_{x \in \mathcal{V}} \max\left(r(x) - q(x), 0\right)
\end{aligned} \tag{9}
$$

and by definition, $P(\text{resampled} = x') = \dfrac{\max(r(x') - q(x'), 0)}{\sum_{x \in \mathcal{V}} \max\left(r(x) - q(x), 0\right)}$. Substituting these into Equation (8), we get:

$$
\begin{aligned}
P(x = x') &= P(\text{proposal } x' \text{ accepted}) \cdot P(\text{proposal} = x') + P(\text{proposal rejected}) \cdot P(\text{resampled} = x') \\
&= \min\left(1, \frac{r(x')}{q(x')}\right) \cdot q(x') + \sum_{x \in \mathcal{V}} \max\left(r(x) - q(x), 0\right) \cdot \frac{\max(r(x') - q(x'), 0)}{\sum_{x \in \mathcal{V}} \max\left(r(x) - q(x), 0\right)} \\
&= \min(q(x'), r(x')) + \max(r(x') - q(x'), 0) \\
&= r(x').
\end{aligned} \tag{10}
$$

The acceptance rate $P(\text{proposal accepted})$ is computed as:

$$
\begin{aligned}
P(\text{proposal accepted}) &= 1 - P(\text{proposal rejected}) \\
&= 1 - \sum_{x \in \mathcal{V}} \max\left(r(x) - q(x), 0\right) \\
&= 1 - \frac{1}{2} \sum_{x \in \mathcal{V}} |r(x) - q(x)| \\
&= 1 - \frac{1}{2} D_{\text{TV}}(r, q).
\end{aligned} \tag{11}
$$

## A.2. Proof of Theorem 3.1

Referring to the proof given by Leviathan et al. (2023), our proof is as follows:

First, after one proposal and one verification, the proposed method generates at least one token, so $P(\#tokens = 0) = 0$ and $P(\#tokens = 1) = 1$. If the model generates $i$ tokens ($1 < i < \gamma$), it means the first $i - 1$ tokens are accepted and the $i + 1$-th token is rejected. Therefore, $P(\#tokens = i) = \alpha^{i-1}(1 - \alpha)$. If the model generates $\gamma$ tokens ($i = \gamma$), it means the first $\gamma - 1$ tokens are accepted. Thus, $P(\#tokens = \gamma) = \alpha^{\gamma-1}$.

$$\mathbb{E}(\#tokens) = \sum_{i=0}^{\gamma} i P(\#tokens = i) = \frac{1 - \alpha^{\gamma}}{1 - \alpha} \tag{12}$$

Note this value differs from that given by Leviathan et al. (2023). This discrepancy is because, in Section 3.2, we do not account for the bonus token.

Assume that the time required to invoke the target model once is $T$, and the time required to invoke a proposal model is $cT$. Therefore, one proposal and one verification together take $(\gamma c + 1)T$ time. The time required to generate one token is $\frac{(1-\alpha)(\gamma c+1)}{1-\alpha^\gamma}T$. In the vanilla collaborative decoding, the time required to generate one token is $(c+1)T$. Therefore, the improvement factor in total walltime is $\frac{(1-\alpha^\gamma)(1+c)}{(1-\alpha)(\gamma c+1)}$.

## B. Algorithm Details

### B.1. Alternate Proposal Framework

We provide the detailed pseudo-code of the alternate proposal framework (see Algorithm 1), which is introduced in Section 3.3.

---

**Algorithm 1** The alternate proposal framework. PROPOSE takes the current token sequence $S$ and a constant $\gamma$ as inputs and generates $\gamma$ tokens $T$. SCORE feeds a sequence to the current verifier to obtain logits $L$ and a bonus token $t$. VERIFY examines $T$ to decide whether it should be accepted according to combined logits.

---

**input** Models $\mathcal{M}_p$, $\mathcal{M}_q$; proposal lengths $\gamma_q, \gamma_p$ ; prefix sequence *prefix*.
  $S \leftarrow prefix$
  $C \leftarrow \emptyset$   ▷ Initialize cached tokens
  **while** not finish **do**
    **if** $C = \emptyset$ **then**
      ▷ Standard speculative decoding step
      proposer $\leftarrow \mathcal{M}_q$
      verifier $\leftarrow \mathcal{M}_p$
      $T \leftarrow$ PROPOSE$(S, \text{proposer}.\gamma)$
    **else**
      ▷ Alternate proposal decoding step
      SWAP(proposer, verifier)
      $T \leftarrow C +$ PROPOSE$(S + C, \text{proposer}.\gamma - C.\text{length})$
    **end if**
    $L, t \leftarrow$ SCORE$(T)$
    $L' = \mathcal{C}'(L)$
    VERIFY$(T, L')$
    **if** all tokens in $T$ are accepted **then**
      $C \leftarrow t$
    **else**
      ▷ Some tokens are rejected, clear $C$ and resample from
      $T \leftarrow$ resample from residual distribution
      $C \leftarrow \emptyset$
    **end if**
    $S \leftarrow S + T$
  **end while**
  **return** $S$

---

### B.2. CoS Framework

The CoS framework is a generalization of the alternate proposal framework to scenarios involving more than three models. Figure 7 illustrates our CoS in a three-model scenario, where the models $\mathcal{M}_1, \mathcal{M}_2, \mathcal{M}_3$ have proposal lengths of $\gamma_1 = 3$, $\gamma_2 = 2$, and $\gamma_3 = 1$, respectively.

Specifically, in step 1, the default proposal model is invoked to generate $\gamma_1 = 3$ proposal tokens: $x_1, x_2, x_3$. In step 2, $\mathcal{M}_2$ is invoked to score $x_1, x_2, x_3$ while naturally generating a bonus token, $x_4$. Since $\mathcal{M}_2$ has a proposal length of $\gamma_2 = 2$, it is

then invoked again to generate an additional $\gamma_2 - 1 = 1$ proposal token to complete its proposal. In step 3, $\mathcal{M}_3$ is called to score $x_1, \ldots, x_5$ in parallel and generate a bonus token, $x_6$.

At this stage, $x_1, x_2, x_3$ have been scored by all models, allowing the combined distributions $r_1(x), r_2(x), r_3(x)$ to be computed and used for verification, as illustrated in step 3. Assuming that $x_1, x_2, x_3$ are all accepted, their corresponding probability distributions are no longer needed and are discarded, as shown in step 4. Subsequently, in step 5, $\mathcal{M}_1$ is invoked again to score $x_4, x_5, x_6$ and generate new proposal tokens: $x_7, x_8, x_9$.

Next, $x_4$ and $x_5$ undergo verification. If $x_4$ is accepted while $x_5$ is rejected, $x_4$ remains unchanged, whereas $x_5$ is replaced with $x_5'$, which is generated through the resampling phase. Since the 5-th token has changed, all subsequent proposal tokens and probability distributions derived from it become invalid and are discarded, as illustrated in step 6. Once these are removed, the default proposal model is invoked to generate $\gamma_1$ new proposal tokens, similar to step 1.

The corresponding pseudocode is provided in Algorithm 2.

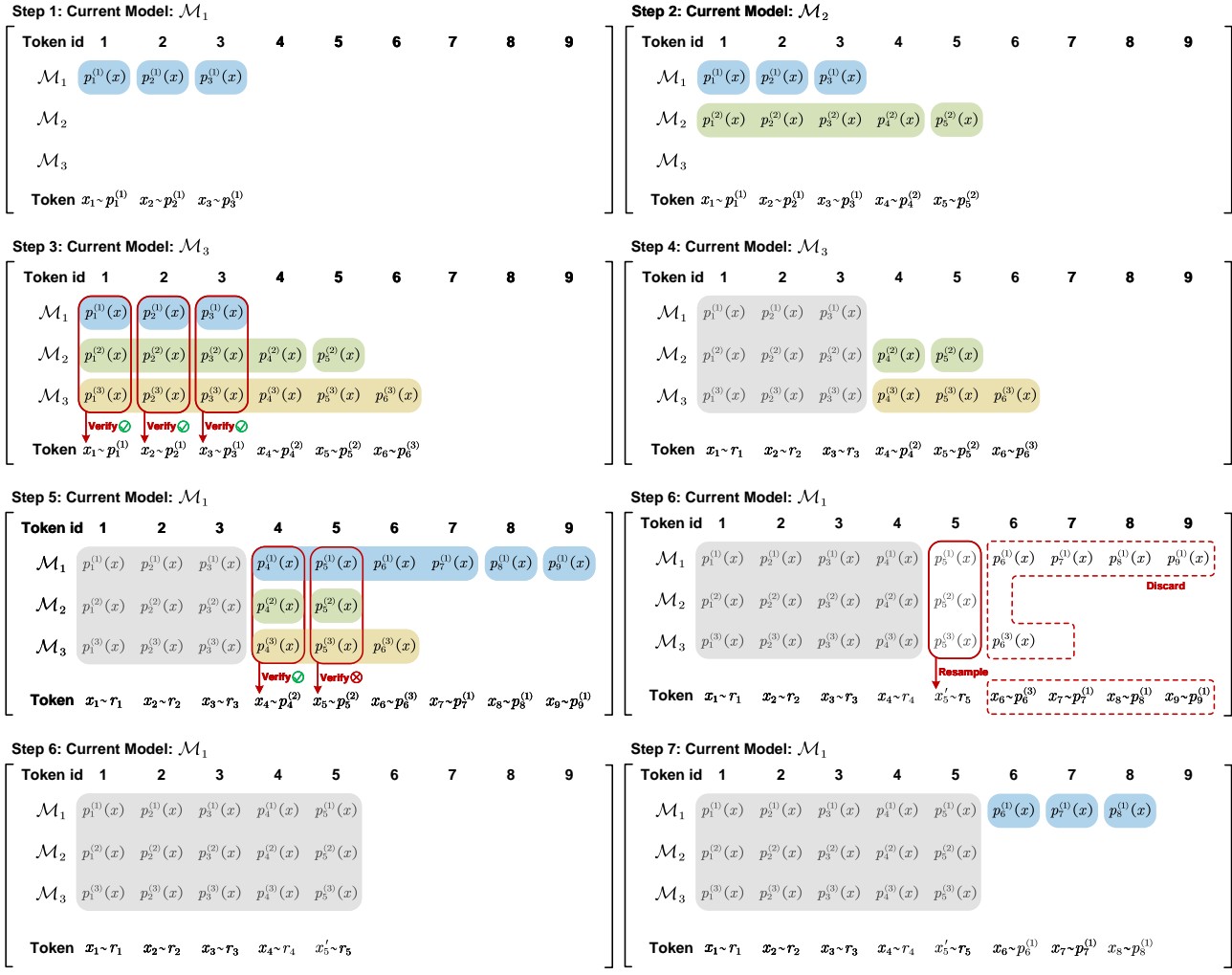

*Figure 7.* The sketch of CoS framework in three-model scenario. The colored boxes represent the stored probability distributions, while the grey boxes represent the cleared ones. Each invocation involves scoring the current proposal tokens and generating a bonus token. The proposal length for model $\mathcal{M}_1, \mathcal{M}_2, \mathcal{M}_3$ is 3, 2, 1, respectively.

---

**Algorithm 2** The CoS framework. PROPOSE employs model $\mathcal{M}$ to take the current token sequence $S$ and a constant $\gamma$ as inputs and generates $\gamma$ tokens $T$. SCORE employs model $\mathcal{M}$ to score a sequence to obtain sequence probabilities $P$, a bonus token $t$ and its probability $p$ . VERIFY examines $T$ to decide whether it should be accepted according to combined probability distribution.

---

**input** Models $\mathcal{M}_1, \ldots, \mathcal{M}_n$; proposal lengths $\gamma_1, \ldots \gamma_n$, prefix sequence *prefix*.
  $S \leftarrow prefix$
  $S_c \leftarrow \emptyset$   ▷ Initialize cached sequence
  $C_i \leftarrow \emptyset$, for $i = 1, \ldots, n$   ▷ Initialize cached probabilities for each model
  **while** not finish **do**
    **if** $S_c = \emptyset$ **then**
      $T, P \leftarrow$ PROPOSE$(\mathcal{M}_1, S, \gamma_1)$   ▷ If no cached sequence, default proposer $\mathcal{M}_1$ is invoked to generate proposal
      $S_c \leftarrow T$   ▷ Cache proposal tokens $T$ and corresponding probabilities $P$
      $C_1 \leftarrow P$
    **else**
      $i \leftarrow \arg\min_i |C_i|$   ▷ Find the model with the shortest cached probabilities, $|\cdot|$ represents the number of elements
      $P, t, p \leftarrow$ SCORE$(\mathcal{M}_i, S_c)$   ▷ Score the $S_c$, generating probabilities of $S_c$, bonus token $t$ and its probability $p$
      $S_c \leftarrow S_c \cup \{t\}$
      $C_i \leftarrow C_i \cup P \cup \{p\}$
      **while** $\forall j, C_j \neq \emptyset$ **do**
        ▷ If all $C_j$ are nonempty, $t_1$ (the first token of $S_c$) must have been scored by all models, so verify it
        $p' \leftarrow \mathcal{C}(p_{11}, \ldots, p_{n1})$   ▷ $p_{ij}$ represents the $j$-th probability of $C_i$
        VERIFY$(p', t_1)$
        **if** $t_1$ is accepted **then**
          $S_c \leftarrow S_c \backslash \{t_1\}$
          $C_j \leftarrow C_j \backslash \{p_{j1}\}$, for $j = 1, \ldots, n$
        **else**
          $t_1 \leftarrow$ resample a token from residual distribution
          $S_c \leftarrow \emptyset$
          $C \leftarrow \emptyset$, for $j = 1, \ldots, n$
        **end if**
        $S \leftarrow S \cup \{t_1\}$
      **end while**
      $T, P \leftarrow$ PROPOSE$(\mathcal{M}_i, S + S_c, \gamma_i - 1)$   ▷ Generate more $\gamma_i - 1$ tokens to finish the proposal
      $S_c \leftarrow S_c \cup T$
      $C_i \leftarrow C_i \cup P$
    **end if**
  **end while**
  **return** $S$

---

# C. Additional Results

## C.1. CoS for Quality-Speed Tradeoff

As outlined in 3.2, creating a weighted ensemble of the proposal and target models in CoS offers a way to balance the tradeoff between quality and speed. We conducted experiments with the Llama-3 model pair on four datasets, adjusting $\lambda$ from 0.1 to 0.9. As shown in Figure 8, increasing $\lambda$ leads to a steady improvement in inference speed but a gradual decline in performance, allowing users to choose a tradeoff that best suits their needs.

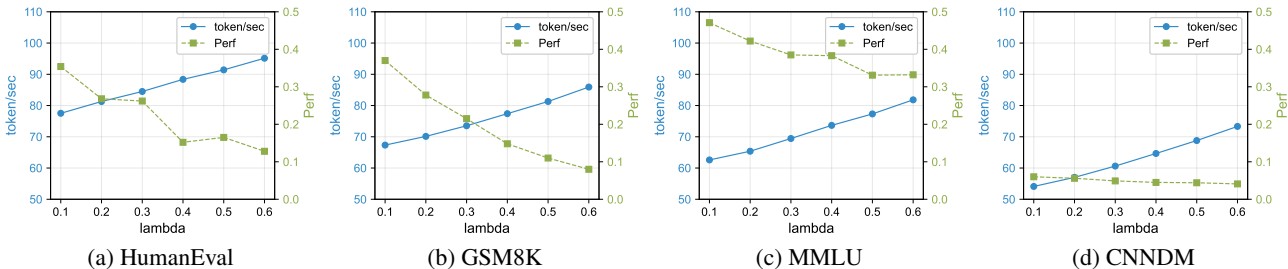

| (a) HumanEval | (b) GSM8K | (c) MMLU | (d) CNNDM |

*Figure 8.* CoS for quality-speed tradeoff

## C.2. Speedup on OPT Model Pair

The results are shown in Table 4.

*Table 4.* The speedup ratio of each method in CD setting.

|  | T | Method | HumanEval | GSM8K | MMLU | CNNDM |
|---|---|---|---|---|---|---|
| | | CD | 1.00x | 1.00x | 1.00x | 1.00x |
| | 0 | CD-SD | 0.97x | 1.05x | 1.47x | 1.40x |
| | | CD-CoS | **3.28x** | **2.61x** | **3.42x** | **3.95x** |
| Opt | | CD | 1.00x | 1.00x | 1.00x | 1.00x |
| | 1 | CD-SD | 1.47x | 1.55x | 2.11x | 1.85x |
| | | CD-CoS | **1.69x** | **1.76x** | **2.16x** | **1.85x** |

## C.3. The Raw Speed

The results are shown in Table 5 and Table 6.

## C.4. Ablation study on Alternate Proposal Framework

The results are shown in Table 7 and Table 8. Note that CD-CoS without APF is equivalent to SCD.

## C.5. Speedup on NPUs

The results are shown in Table 9 and Table 10.

*Table 5.* The raw speed of each method under WE setting. The table reports the average number of tokens generated per second. Models are of comparable sizes.

| Model Pair | Method | HumanEval | GSM8K | MMLU | CNNDM |
|---|---|---|---|---|---|
| | WE | 22.617 | 22.054 | 20.459 | 20.782 |
| Llama-Vicuna | WE-SD | 28.723 | 26.685 | 24.346 | 23.899 |
| | WE-CoS | **35.734** | **33.522** | **28.847** | **30.341** |
| | WE | 49.624 | 49.025 | 47.764 | 46.966 |
| Qwen-3b | WE-SD | 56.075 | 51.966 | 52.062 | 50.723 |
| | WE-CoS | **80.390** | **74.518** | **67.824** | **64.813** |
| | WE | 82.584 | 77.941 | 79.631 | 76.625 |
| Qwen-1.5b | WE-SD | 91.668 | 88.073 | 86.001 | 84.287 |
| | WE-CoS | **128.831** | **113.793** | **106.705** | **103.443** |
| | WE | 57.048 | 53.869 | 53.742 | 56.286 |
| Qwen-1.5b | WE-SD | 55.050 | 51.082 | 52.666 | 51.927 |
| | WE-CoS | **105.485** | **68.287** | **74.426** | **85.980** |

*Table 6.* The raw speed of each method under CD setting. Models are of different sizes.

| Model Pair | T | Method | HumanEval | GSM8K | MMLU | CNNDM |
|---|---|---|---|---|---|---|
| | | CD | 39.387 | 38.735 | 38.969 | 38.100 |
| | 0 | CD-SD | 80.349 | 70.110 | 59.233 | 60.198 |
| | | CD-CoS | **87.833** | **77.470** | **68.975** | **61.341** |
| Llama-3 | | CD | 38.981 | 38.585 | 39.028 | 38.124 |
| | 1 | CD-SD | 60.421 | 46.688 | 46.834 | 40.793 |
| | | CD-CoS | **64.319** | **55.562** | **51.127** | **44.986** |
| | | CD | 49.314 | 48.129 | 49.700 | 46.215 |
| | 0 | CD-SD | 56.711 | 77.969 | 53.676 | 42.980 |
| | | CD-CoS | **62.136** | **79.413** | **83.496** | **60.080** |
| Llama-2 | | CD | 47.325 | 47.015 | 47.492 | 43.512 |
| | 1 | CD-SD | 44.486 | 54.537 | 58.415 | 47.863 |
| | | CD-CoS | **54.424** | **56.418** | **65.064** | **48.298** |
| | | CD | 23.525 | 23.525 | 19.576 | 19.692 |
| | 0 | CD-SD | 23.211 | 24.701 | 28.776 | 27.568 |
| | | CD-CoS | **78.487** | **61.400** | **66.949** | **77.783** |
| OPT | | CD | 23.934 | 23.422 | 19.492 | 19.671 |
| | 1 | CD-SD | 35.182 | 36.304 | 41.128 | 36.391 |
| | | CD-CoS | **40.448** | **41.222** | **42.102** | **36.391** |

*Table 7.* Ablation study on alternate proposal framework (APF) under weighted ensemble (WE) setting ($\lambda = 0.5$). Models are of comparable sizes.

| Model Pair | Method | HumanEval | GSM8K |
|---|---|---|---|
| Llama-Vicuna | w/o APF | 1.46x | 1.39x |
| | WE-CoS | **1.58x** | **1.52x** |
| Qwen-3b | w/o APF | 1.40x | 1.32x |
| | WE-CoS | **1.62x** | **1.52x** |

*Table 8.* Ablation study on alternate proposal framework (APF) under contrastive decoding (CD) setting ($\mu = 0.1$). Models are of different sizes.

| Model Pair | T | Method | HumanEval | GSM8K |
|---|---|---|---|---|
| Llama-3 | 0 | w/o APF | 1.98x | 1.72x |
| | | CD-CoS | **2.23x** | **2.00x** |
| | 1 | w/o APF | 1.41x | 1.24x |
| | | CD-CoS | **1.65x** | **1.44x** |
| Llama-2 | 0 | w/o APF | 1.18x | 1.59x |
| | | CD-CoS | **1.26x** | **1.65x** |
| | 1 | w/o APF | 0.89x | 1.21x |
| | | CD-CoS | **1.15x** | **1.20x** |

*Table 9.* The raw speed of each method under both WE and CD settings using the Ascend 910B3 NPU.

| Model Pair | Method | HumanEval | GSM8K | MMLU | CNNDM |
|---|---|---|---|---|---|
| Llama-Vicuna | WE | 7.60 | 5.83 | 2.60 | 2.35 |
| | WE-SD | 13.59 | 8.73 | 3.81 | 3.55 |
| | WE-CoS | **14.14** | **9.98** | **4.32** | **4.06** |
| Llama-2 | CD | 11.98 | 10.63 | 4.96 | 4.51 |
| | CD-SD | 13.00 | 16.28 | 7.098 | 4.83 |
| | CD-CoS | **16.71** | **16.30** | **7.54** | **5.23** |

*Table 10.* The speedup ratio of each method under both WE and CD settings using the Ascend 910B3 NPU.

| Model Pair | Method | HumanEval | GSM8K | MMLU | CNNDM |
|---|---|---|---|---|---|
| Llama-Vicuna | WE | 1.00x | 1.00x | 1.00x | 1.00x |
| | WE-SD | 1.79x | 1.50x | 1.47x | 1.51x |
| | WE-CoS | **1.86x** | **1.71x** | **1.66x** | **1.73x** |
| Llama-2 | CD | 1.00x | 1.00x | 1.00x | 1.00x |
| | CD-SD | 1.09x | 1.53x | 1.43x | 1.07x |
| | CD-CoS | **1.39x** | **1.53x** | **1.52x** | **1.16x** |

