# OpenReview forum: "Fast Large Language Model Collaborative Decoding via Speculation"
_ICML.cc/2025/Conference — ICML 2025 poster_

### Official Review · Reviewer_3eWy · 2025-03-13

**Overall Recommendation:** 4

**Summary:**

This paper introduces "Speculative Ensemble" (SE), a novel framework that accelerates Large Language Model (LLM) ensembles without sacrificing performance. While ensemble methods enhance LLMs by combining multiple models, they suffer from high computational costs. The authors build on speculative decoding—where a small model generates tokens sequentially and a larger model verifies them in parallel—with two key insights: (1) the verification distribution can be the ensemble distribution of both models, and (2) alternating each model as proposer and verifier enhances efficiency. The approach generalizes to n-model ensembles and theoretical analysis proves SE is never slower than standard ensembles. Experiments across various tasks demonstrate speed improvements of 1.11x–2.23x over standard ensemble techniques without compromising generation quality.

**Claims And Evidence:**

The paper's claims are generally well-supported by both theoretical analysis and empirical evidence. They prove and experimentally show that the speculative ensemble improves speed without sacrificing performance, and it is not slower than an ensemble approach.

One main concern I have is that based on practical intuition, the approach does not seem to have a clear speedup over the normal ensemble. In the case of 2 models alternating, in Figure 2, at best, each model processes their tokens, and then they are verified by another model. If they are close in size, the expectation is that the parallel run of these models (normal ensemble) will be equal or better than the current proposal. This is because, in the current proposal, we have the autoregressive steps at every step (except for verification), which can run the models in parallel, and avoid extra time for verification. If the models differ in size greatly, then in half of the steps, we are doing better than the parallel ensemble, and in the other half we are not. Hence, it is not clear where the speedup is coming from.

**Essential References Not Discussed:**

Not that I am aware of.

**Experimental Designs Or Analyses:**

The experimental design in the paper is generally sound, with a comprehensive evaluation across multiple model architectures, tasks, and ensemble functions. However, the concern mentioned above, should be addressed in some experimental results, by showcasing the profile of each run (parallel ensemble, and SD ensemble), so we can have a better understanding of how the model achieve this speedup.

**Methods And Evaluation Criteria:**

The paper uses appropriate methodologies and evaluation criteria for assessing the proposed Speculative Ensemble framework. The authors evaluate their approach across diverse tasks (code generation, mathematical reasoning, multi-task understanding, and text summarization) using established benchmarks (HumanEval, GSM8K, MMLU, and CNNDM), which effectively represent a range of LLM applications. Their evaluation metrics focus on tokens generated per second and speedup ratios relative to standard ensemble methods. The experimental design includes various model pairs (Llama-Vicuna, Qwen, Llama-3, OPT), testing both two-model and three-model configurations, and comparing multiple ensemble functions (weighted ensemble and contrastive decoding). Additionally, the comprehensive ablation studies examining the impact of various parameters (λ, μ, and γ) further strengthen the evaluation by providing insights into the factors affecting performance.

**Other Comments Or Suggestions:**

NA

**Other Strengths And Weaknesses:**

mentioned above.

**Questions For Authors:**

mentioned above.

**Relation To Broader Scientific Literature:**

The proposal is an interesting approach for speeding up the ensemble methods, and can be useful in various application.

**Theoretical Claims:**

The paper provides well-structured and logical proofs for its key theoretical claims. The authors offer detailed proofs for the correctness of speculative-based ensemble (showing tokens align with the ensemble distribution), acceptance rate calculation, and the speed improvement factors.

---

> ### Author Rebuttal · Authors · 2025-03-31
>
> We thank the reviewer for their time and insightful comments. Below, we address each concern in detail.
>
> **Claims And Evidence 1: compare to PE when model sizes are close**
>
> First, we evaluate the speedup of *parallel ensemble* (PE). However, PE is even slower than the sequential ensemble. For details on the experimental setup, results, and further discussion, please refer to our response to Reviewer hXqY under 'Other Comments or Suggestions'.
>
> Moreover, as you noted, when model sizes are comparable, SE is theoretically expected to perform similarly to or slower than PE. Nonetheless, we maintain that SE remains a promising and broadly applicable approach. Our reasoning is as follows:
>
> 1. In practice, PE is limited by its reliance on complex engineering and powerful hardware. It also requires twice the throughput and more GPU memory. Consequently, sequential ensemble is also widely used, as seen in repositories like xydaytoy/EVA, starrYYxuan/UniTE, and cmavro/PackLLM. SE can greatly speedup sequential ensemble without requiring specialized engineering or additional throughput and GPU memory, highlighting its practical potential.
>
> 2. Furthermore, if SE is equipped with the same throughput as PE (i.e., double its current throughput), it can be combined with PE to accelerate more models. For example, the 2-model ensemble in Figure 2 can be naturally extended to a 4-model ensemble. The procedural steps for this extension are detailed below, and a schematic illustration is provided in [SE with_PE](https://anonymous.4open.science/r/SE-Rebuttal-Supplement-ICML25-667C/se_with_pe.pdf).
>
>    - Step 1: Models $\mathcal{M}_1$ and $\mathcal{M}_2$ are invoked in parallel to produce distributions $p_1^{(1)}(\cdot)$ and $p_1^{(2)}(\cdot)$. A token $x_1^{(1)}$ is sampled from $p_1^{(1)}(\cdot)$ as the proposal, and its score under $\mathcal{M}_2$, given by $p_1^{(2)}(x_1^{(1)})$, is saved.
>    - Step 2: Similarly, $\mathcal{M}_3$ and $\mathcal{M}_4$ are invoked in parallel to score $x_1^{(1)}$ and generate bonus distributions $p_2^{(3)}(\cdot)$ and $p_2^{(4)}(\cdot)$. A bonus token $x_2^{(3)}$ is sampled from $p_2^{(3)}(\cdot)$, and its score under $\mathcal{M}_4$, i.e., $p_2^{(4)}(x_2^{(3)})$, is saved.
>    - At this stage, $x_1^{(1)}$ has been scored by all 4 models and proceeds to verification. If accepted, $x_1^{(1)}$ is treated as a sample from the ensemble distribution, as illustrated in Step 3.
>    - Step 4: The process is repeated with $\mathcal{M}_1$ and $\mathcal{M}_2$ invoked again to score $x_2^{(3)}$ and generate new bonus distributions, mirroring the procedure in Step 2.
>
>    The algorithm retains the properties discussed in the paper, including losslessness, and it is never slower than the 4-model PE.
>
>    Note that the method described here is a preliminary version and can be further optimized. For instance, in Step 1, we can sample two tokens from both  $p_1^{(1)}(\cdot)$ and $p_1^{(2)}(\cdot)$, and verify them in parallel using a tree attention to further improve the acceptance rate. We will explore this in future work to further enhance the performance of SE.
>
> **Claims And Evidence 2: compare to PE when model sizes differ**
>
> When the model sizes differ greatly, speedup is mainly coming from reducing the number of large model invocations.
>
> In the optimal case, PE reduces only the computational time of the small model, while the large model remains a major time bottleneck. In contrast, SE—under setting shown in Figure 2—can halve the number of large model invocations, resulting in a substantially more effective speedup.
>
> On the other hand, the speedup comes from allowing the smaller model focuses on sequential generation, while the larger model focuses on parallel verification. This shares a common acceleration principle with SD.
>
> As noted on Page 5, Line 251 (left column), two hyperparameters, $\gamma_p$ and $\gamma_q$, control the proposal length for each model when acting as the proposer. When model sizes differ greatly, the larger model’s $\gamma$ is set to 1, and the smaller model is assigned a higher value (e.g., 5), as described in Section 4.1, under “Configuration of $\gamma$”. As a result, the smaller model is invoked more frequently—about 5 times as often as the larger model—rather than "half" of the total invocations. This setup ensures that the smaller model focuses on sequential proposal generation, while the larger model conducts parallel verification, echoing the acceleration strategy of vanilla SD. This is further supported by the ablation study in Figure 6(b), which demonstrates that speedup consistently increases with higher $\gamma$ values of the small model.
>
> **Experimental Designs Or Analyses:**
>
> We also provide a case to better illustrate how SE achieves greater speedup compared to SD; please see our response to Reviewer hXqY under "Questions for Authors".
>
> If you have any questions or concerns, please let us know. We’re committed to addressing them.

---

> > ### Comment · Reviewer_3eWy · 2025-04-04
> >
> > Thanks to the authors for their response. I have raised my score, as my concerns are mainly addressed. This paper can be a good application venue for speculative decoding.

---

> > > ### Author Response · Authors · 2025-04-05
> > >
> > > Thank you for the thoughtful feedback and the improved evaluation. We’re glad that our responses addressed your concerns and that you see the paper’s potential as a good application venue for speculative decoding. We will integrate the rebuttal points into the revision to further improve the paper.

---

### Official Review · Reviewer_m8L7 · 2025-03-15

**Overall Recommendation:** 2

**Summary:**

The authors extend speculative decoding to ensemble models and demonstrate, through both theoretical analysis and empirical results, that their approach outperforms standard ensemble baselines.

**Claims And Evidence:**

Yes

**Essential References Not Discussed:**

Not that I'm aware of.

**Experimental Designs Or Analyses:**

1. Do you test larger models, larger than 7B?
2. Could you design experiments to compare with more baselines/related works? Only comparing with pure ensemble models and the original speculative decoding seems insufficient to fully evaluate the contribution of the proposed method.

**Methods And Evaluation Criteria:**

Yes

**Other Comments Or Suggestions:**

No minor issues.

**Other Strengths And Weaknesses:**

Strengths:
1. The proposed method shows improvements over standard ensemble models, demonstrating its practical value within that setting.

Weaknesses:
1. The method appears to be a relatively minor adaptation of standard speculative decoding applied specifically to ensemble models, which raises concerns about its general applicability beyond this narrow use case.
2. The theoretical analysis closely follows prior speculative decoding work and does not introduce significantly novel insights.
3. It is unclear whether the authors have compared their approach against strong non-ensemble baselines. A comparison in terms of speedup and accuracy trade-offs with single-model speculative decoding would help contextualize the contribution.
4. The paper includes limited baseline comparisons, making it difficult to assess the overall effectiveness and competitiveness of the proposed method.

**Questions For Authors:**

1. Could you compare the speedup and accuracy trade-offs with single-model speculative decoding, not just with ensemble models?
2. Have you tested on larger models? How does the model size affect the results?
3. How does the method compare with batch speculative decoding methods such as Medusa [1]?
4. Could you design experiments to compare with more baselines/related works? Only comparing with pure ensemble models and the original speculative decoding seems insufficient to fully evaluate the contribution of the proposed method.

[1] Cai, Tianle, et al. "Medusa: Simple llm inference acceleration framework with multiple decoding heads." arXiv preprint arXiv:2401.10774 (2024).

**Relation To Broader Scientific Literature:**

Speculative decoding is important for efficient AI.

**Theoretical Claims:**

The theoretical analysis largely follows the original speculative decoding framework, but appears to be correct.

---

> ### Author Rebuttal · Authors · 2025-03-31
>
> We thank the reviewer for their time and insightful comments. Below, we address each concern in detail.
>
> For clarity and brevity, we use the following abbreviations: Experimental Designs Or Analyses (EDOA), Weaknesses (W), and Questions (Q).
>
> **W1: Speculative Ensemble (SE) offers two non-trivial improvements over vanilla SD.**
>
> Compared to simply adapting vanilla SD to ensemble scenarios, SE offers two non-trivial improvements:
>
> 1. Compared to vanilla SD, where one model consistently serves as the proposer and the other as the verifier, we introduce an alternating proposal framework (APF) in Section 3.3, as shown in Figure 2. This method switches the roles of proposer and verifier during decoding, and is specifically designed to enhance efficiency in ensemble settings. We also validate its effectiveness through ablation experiments shown in [Ablation of APF](https://anonymous.4open.science/r/SE-Rebuttal-Supplement-ICML25-667C/ablation_of_apf.md). For a case-specific explanation, please refer to our response to reviewer hXqY under 'Questions for Authors'.
> 3. Unlike the vanilla SD, which employs two models, the proposed SE extends to an n-model scenario to accelerate the n-model ensemble in Section 3.4, as shown in Figure 3 and Figure 7.
>
> Moreover, Reviewer hXqY described our method as “clever, novel, and well-validated,” while Reviewer 3eWy referred to it as a “novel framework,” both highlighting the non-trivial improvements offered by our approach.
>
> **W1: concerns about general applicability**
>
> Our proposed SE not only accelerates traditional weighted ensemble (WE) methods (Eq 4), but also accelerates ensembles of any form, size, or number of LLMs at the probability or logits level. This includes techniques such as contrastive decoding (CD) [1] (Eq 5) and decoding-time realignment [2], demonstrating its broad applicability. As Reviewer 3eWy noted, SE “can be useful in various application,” further supporting this point.
>
> **W2: concerns about theoretical novel insights**
>
> While our theoretical analysis of correctness (i.e., losslessness) and speedup follows original SD paper, we introduce two novel theoretical insights tailored to the ensemble scenario:
>
> 1. As discussed on page 4, line 205 (left column), in vanilla SD, estimating the expected speedup requires extensive experiments to estimate the acceptance rate $\alpha$. However, in WE setting, the parameter $\lambda$ (Eq 4) naturally serves as a lower bound for $\alpha$. This allows us to estimate the speedup in advance and choose a suitable proposal model, such as those described in Corollary 3.5.
> 2. Vanilla SD does not guarantee acceleration; in fact, it may result in slower speed when $\alpha$ is small. In contrast, SE is theoretically proven to be no slower than the standard ensemble, even in the worst case, as established in Corollary 3.7.
>
> **Q1, 3, 4 & W3, 4: compare with more baselines**
>
> Thank you for your valuable suggestions. To better demonstrate SE, we conducted experiments comparing SE with three non-ensemble baselines: Large Model, vanilla SD, and Medusa. The results are shown in [Non-ensemble](https://anonymous.4open.science/r/SE-Rebuttal-Supplement-ICML25-667C/non_ensemble.md). Note that for Medusa, we only reported the results for Vicuna, because it only provides the pretrained draft model for Vicuna. For Qwen-3B, we did not report results for vanilla SD, because there is no suitable draft model for it.
>
> Additionally, it is important to note that SE, as an acceleration algorithm, is a line of research parallel to SD. It does not focus on finding a quality-speed trade-off. Instead, SE focus on accelerates inference while preserving the performance gains provided by ensemble.
>
> **EDOA1 & Q2: results for models larger than 7B and the impact of model size**
>
> As described in Section 4.1, "Model Pair Configuration" and in Table 1, we evaluated the CD setting using the model pair (OPT-13B, OPT-125m). The corresponding results are presented in Table 4.
>
> In addition, we also tested a larger model pair (Llama-2-13B, Vicuna-13B) under the WE setting. The results are shown in [Larger Model](https://anonymous.4open.science/r/SE-Rebuttal-Supplement-ICML25-667C/larger_model.md).
>
> Together with the results in Tables 2 and 3, we observe that SE achieves greater speedup as model size increases in both the WE and CD scenarios, with a more pronounced effect in CD. In WE, this trend may be due to improved model performance with larger sizes, which increases similarity between models and leads to a higher acceptance rate. In the CD scenario, the growing speedup may stem from the increasing cost of invoking the larger model.
>
>
>
> If you have any further questions or concerns, please feel free to let us know. We are committed to addressing any concerns to the best of our ability.
>
> [1] Contrastive Decoding: Open-ended Text Generation as Optimization
>
> [2] Decoding-time Realignment of Language Models

---

> > ### Comment · Reviewer_m8L7 · 2025-04-02
> >
> > Thank you for the rebuttal. Based on the non-ensemble comparisons, where the proposed method is significantly slower than standard speculative decoding (SD), I remain unconvinced that applying speculative decoding to weighted ensembles offers broadly applicable benefits. Therefore, I am inclined to maintain my original score.

---

> > > ### Author Response · Authors · 2025-04-02
> > >
> > > Thank you for your response. We think that several key points in our paper still require clarification.
> > >
> > > **1. Standard SD and SE achieve different types of acceleration and are not comparable**
> > >
> > > Take the Llama-Vicuna model pair as an example. Standard SD accelerates a **single 7B model**, whereas the WE-SE results are based on an ensemble of **two 7B models**. Since a single model naturally runs faster than an ensemble, directly comparing their speeds is inappropriate.
> > >
> > > A more appropriate comparison is between WE-SE and WE-SD—the latter applies SD directly to the ensemble, as described in Section 4.1, *“Ensemble functions and methods.”* As shown in Table 2, WE-SE achieves significantly greater speed than WE-SD.
> > >
> > > **2. The benefits of applying SD or SE to weighted ensembles**
> > >
> > > As shown in our [Raw Speed](https://anonymous.4open.science/r/SE-Rebuttal-Supplement-ICML25-667C/raw_speed.md) results, if someone wants to ensemble Llama and Vicuna, the standard ensemble can only achieve a speed of only 22.617 tokens/sec. In contrast, applying SD (WE-SD) increases the speed to 28.723 tokens/sec, and applying SE further boosts it to 35.734 tokens/sec, resulting in a substantial efficiency improvement.
> > >
> > > Additionally, it is important to note that **single-model SD approaches can not accelerate the ensemble**.

---

### Official Review · Reviewer_wgdK · 2025-03-19

**Overall Recommendation:** 3

**Summary:**

## Update after Rebuttal

My concern regarding the insufficient analysis of the quality–speedup trade-off was addressed by the authors’ rebuttal, therefore I have reflected this by increasing my score from 2 to 3, i.e., leaning towards acceptance.

However, the results of comparing with non-ensemble baselines suggest that the proposed method is ~$2\times$
 slower than standard speculative decoding (as also noted by Reviewer m8L7). This prevents me from giving a higher rating.

IMHO, this actually raises concerns about the necessity of using the ensemble approach --- when comparable or even better performance might be achievable by pairing a small and a large model (e.g., 1B–14B in standard speculative decoding), using two models of similar size as in the ensemble setup (e.g., 7B–7B) seems unnecessary since the non-ensemble methods could potentially lead to both faster generation speed and better generation quality.

On a related note, this ICLR 2025 paper (SpecCascade [1]) appears to be a closely relevant work, which is able to provide better cost-quality trade-offs than their sequential cascade and speculative decoding counterparts. **Given the similarity between Speculative Ensemble and SpecCascade, it is suggested that the authors discuss this work and better position the unique contribution of the submission.**

[1]. Faster Cascades via Speculative Decoding, ICLR 2025.

## Original Summary

This paper introduces Speculative Ensemble (SE), a method designed to accelerate large language model (LLM) ensembles by leveraging speculative decoding principles. Instead of computing ensemble predictions independently for each model, SE employs a verification scheme where models alternate as proposers and verifiers. The paper claims that this design maintains ensemble quality compared to standard ensemble methods while achieving faster decoding speeds. Theoretical analysis demonstrates that the proposed SE is guaranteed to be no slower than a standard ensemble under certain assumptions. Empirical results on benchmarks such as HumanEval, GSM8K, MMLU, and CNNDM show speedups ranging from 1.11x to 2.23x over standard ensembles. However, no comparison of generation quality is provided.

**Claims And Evidence:**

- The claim that Speculative Ensemble (SE) maintains performance compared to standard ensembles lacks empirical validation. In Section 4, the experiments primarily focus on speedup comparisons, while no quality metrics (e.g., accuracy) are provided to confirm that speculative ensembling does not degrade generation performance. As a result, it remains unclear whether SE preserves the benefits of ensembling.

**Essential References Not Discussed:**

N/A

**Experimental Designs Or Analyses:**

- The experiments demonstrate speedup but fail to evaluate generation quality.
- The quality-speed tradeoff study in Appendix C.1 is insufficient. No ablations are provided to separate the impact of speculative decoding vs. ensemble benefits.
- The paper does not compare SE against simpler baselines, such as using the large model alone or using a standard speculative decoding setup with one small drafter and one large verifier (w/o alternating the drafter and verifier).

**Methods And Evaluation Criteria:**

- Lack of generation quality evaluations. While the speedup metric (tokens per second) is reasonable, the absence of generation quality evaluations is a significant concern. As stated in Corollary 3.5, there exists a hyperparameter $\lambda$ such that the proposed Speculative Ensemble (SE) is guaranteed to be at least as fast as a standard ensemble. However, the choice of $\lambda$ inherently impacts ensemble quality. Although certain values of $\lambda$ may lead to speedups, they could also result in performance degradation, potentially making SE less effective than non-ensemble methods or standard ensemble methods. Without empirical evidence demonstrating that SE improves generation quality, the practical utility of the method remains uncertain.

- Insufficient baseline comparisons. The authors do not compare SE against non-ensemble baselines to demonstrate its effectiveness in terms of generation speedup and quality. A basic evaluation setup should include the following typical baselines: 1) single-model baselines: using only the large or smaller model alone for token generation. 2) standard speculative decoding: employing a fixed small model as the drafter and a large model as the verifier for token generation.

Without such comparisons, it is difficult to assess whether SE provides meaningful improvements over simpler or more established approaches.

**Other Comments Or Suggestions:**

N/A

**Other Strengths And Weaknesses:**

Other Strengths
- An interesting attempt to extend speculative decoding to ensemble settings

Other Weaknesses
- Missing comparisons with simpler baselines
- No empirical validation of quality preservation—only speedup is measured
- Alternating proposer/verifier is not clearly justified—it may degrade efficiency and quality

**Questions For Authors:**

1. How does Speculative Ensemble (SE) impact generation quality compared to baseline methods?
2. The ensemble setup seems less practical. While pairing a smaller model with a large model is common in speculative decoding, ensembling is typically used with comparable models to achieve further performance gains. In what scenarios would it be necessary to ensemble a large model with a much smaller model? How does SE compare to simply using a single large model? Is ensembling actually necessary for performance gains in the setting described in this paper?
3. Can the authors justify why alternating between proposer and verifier roles improves efficiency, rather than using the large model exclusively for verification? This finding appears to depend on the specific choice of drafter and verifier models. Would the same efficiency gains hold if the drafter and verifier models were of comparable sizes?

**Relation To Broader Scientific Literature:**

The paper borrows principles from speculative decoding and applies them to LLM ensembles.

**Theoretical Claims:**

The mathematical proofs seem correct, but they do not address whether SE actually improves generation quality.

---

> ### Author Rebuttal · Authors · 2025-03-31
>
> We thank the reviewer for their time and insightful comments. Below, we address each concern in detail.
>
> For brevity, we use the following abbreviations: Claims and Evidence (CAE), Methods and Evaluation Criteria (MAEC), Theoretical Claims (TC), Experimental Designs or Analyses (EDOA), Weaknesses (W), and Questions for Authors (Q).
>
> **CAE & MAEC1 & TC & EDOA1 & W2 & Q1: concerns about generation quality**
>
> We apologize for the lack of clarity regarding generation quality. In the SD domain, performance is theoretically well-established, so researchers typically focus on comparing speed rather than performance [1] [2] [3]. Our approach follows this standard practice.
>
> From both theoretical and experimental perspectives, we confirm that SE consistently maintains ensemble quality.
>
> Theoretically, as shown in Appendix A.1, we proved that the generated tokens precisely follow the ensemble distribution—that is, they can be regarded as samples from this distribution. Reviewers hXqY, A7ww, and 3eWy also endorsed the correctness of our proof.
>
> To further validate this, we conducted additional experiments. As shown in [Generation Quality](https://anonymous.4open.science/r/SE-Rebuttal-Supplement-ICML25-667C/generation_quality.md), when T=0, the performance is exactly the same (as proved in Section 3.2 of [4]), while when T=1, due to randomness, the performance shows slight differences but remain largely consistent. For the weighted ensemble (WE), we report only T=1 case, as T=0 is uncommon in WE setting, as discussed in Section 4.1 "Ensemble functions and methods".
>
> **MAEC1 : concern about Corollary 3.5**
>
> Corollary 3.5 states that for any given $\lambda$, there exists a $\gamma$ that guarantees an ensemble speedup. Notably, $\lambda$ is not a hyperparameter of the SE algorithm; instead, the proposal length $\gamma$ is. As an acceleration method, SE aims to speed up the ensemble under a given $\lambda$. The selection of $\lambda$ follows the same procedure as in standard ensembles—that is, it is set to the value that yields the best performance. For example, [5], [6], and [7] respectively set $\lambda$ based on the trade-off between model purification and standard performance, the performance on a development set, and the perplexity.
>
> In addition to accelerating common weighted ensembles (Equation (4)), the proposed SE can also speed up any form of probability- or logits-level ensembles (Equation (3)). In this more general setting, Corollary 3.7 provides a theoretical guarantee of SE's acceleration.
>
> **MAEC2 & EDOA3 & W1: compare with non-ensemble baselines**
>
> Regarding this, please refer to our response to Reviewer m8L7 under "Q1, 3, 4 & W3, 4".
>
> **Q2: the practicality of ensemble small and large models**
>
> Regarding this question, please refer to our response to Reviewer A7ww under "MAEC1".
>
> **EDOA2: concerns about ablations**
>
> First, the quality-speed tradeoff in Appendix C.1 is not a central focus of the paper, please refer to our response to Reviewer A7ww under "MAEC2" for a more detailed explanation.
>
> Second, we did not separate the impact of speculative decoding and ensemble benefits because SE is designed specifically for ensemble scenarios, rather than treating ensemble as a separate component. While it is well known that ensemble can enhance LLM performance, it typically incurs slow inference. SE aims to accelerate inference while preserving these performance gains.
>
> **W3 & Q3: concerns about alternate proposal**
>
> First, please refer to our response to Reviewer hXqY under "Questions For Authors" for an example illustrating the impact of using the alternate proposal framework (APF). As shown, APF enables the generation of an additional bonus token within the same number of model invocation, thereby enhancing efficiency.
>
> Second, our ablation study confirms the effectiveness of APF. The results are shown in [Ablation of APF](https://anonymous.4open.science/r/SE-Rebuttal-Supplement-ICML25-667C/ablation_of_apf.md).
>
> Third, as detailed in Section 4.1, "Ensemble Functions and Methods" and Table 1, we evaluated 4 groups of comparable model configurations (e.g., Llama-2-7B + Vicuna-7B). The results in Table 2 show that APF can still enhance efficiency if the drafter and verifier models are of comparable sizes.
>
>
>
> If you have any further questions or concerns, please feel free to let us know. We are committed to addressing any concerns to the best of our ability.
>
>
>
> [1] Fast Inference from Transformers via Speculative Decoding
>
> [2] EAGLE: Speculative Sampling Requires Rethinking Feature Uncertainty
>
> [3] GliDe with a CaPE: A Low-Hassle Method to Accelerate Speculative Decoding
>
> [4] Speculative Decoding: Exploiting Speculative Execution for Accelerating Seq2seq Generation
>
> [5] Purifying Large Language Models by Ensembling a Small Language Model
>
> [6] Ensemble Learning for Heterogeneous Large Language Models with Deep Parallel Collaboration.
>
> [7] Relative representations enable zero-shot latent space communication

---

> > ### Comment · Reviewer_wgdK · 2025-04-05
> >
> > I thank the authors for answering my questions and providing additional experimental results.
> >
> > Regarding the concern about generation quality, I was referring to the fact that the hyperparameters $\lambda$ (for WE) and $\mu$ (for CD) can affect the effectiveness of the methods.
> >
> > As shown in Figure 8, to achieve higher speedups in WE-SE and CD-SE, $\lambda$ and $\mu$ need to be carefully chosen. **However, the optimal configuration of $\lambda$ and $\mu$ for WE-SE and CD-SE might not be the same as that for WE and CD.** I understand that speculative decoding maintains output consistency, but for ensemble methods, the selection of $\lambda$ and $\mu$ that optimizes ensemble performance may not lead to the best speedup for WE-SE and CD-SE. So there could be a trade-off.
> >
> > For example, under the current setting of $\lambda=0.5$ and $\mu=0.1$, WE-SE and CD-SE can maintain the same generation quality as WE and CD (since they share the same hyperparameters) while achieving faster generation. However, the optimal performance of WE and CD might occur under a different hyperparameter choice (say, $\lambda=0.9$ and $\mu=0.3$), under which WE-SE and CD-SE might not achieve meaningful speedups.
> >
> > Therefore, I think my question is: **when WE and CD use their own optimal $\lambda$ and $\mu$ values (which may differ from the optimal configuration for WE-SE and CD-SE since they need to balance the generation quality and speedup), what is the resulting generation quality of WE-SE and CD-SE compared to WE and CD?**

---

> > > ### Author Response · Authors · 2025-04-06
> > >
> > > We apologize for the earlier misunderstanding regarding your concern about generation quality and appreciate your thoughtful follow-up. We remain committed to addressing your remaining concerns with clarity and thoroughness.
> > >
> > > **1. The choice of $\lambda$ and $\mu$ in SE**
> > >
> > > In the proposed SE, the parameters $\lambda$ and $\mu$ are not chosen to balance ensemble performance and speedup. Instead, we recommend the following strategy for using SE:
> > >
> > > 1. First, identify the optimal value of $\lambda$ or $\mu$ in the standard ensemble—that is, the value that yields the greatest performance improvement.
> > > 2. Then, using this optimal $\lambda$ or $\mu$, apply SE to further accelerate the ensemble while preserving this optimal performance gain.
> > >
> > > Under this selection strategy, high generation quality is well ensured. However, as you correctly noted, SE may not achieve the optimal acceleration. Nonetheless, our ablation study demonstrates that SE still provides a substantial acceleration.
> > >
> > > In WE setting, as discussed in Section 4.3 "Speedup ratio for different weight $\lambda$ in WE" and illustrated in Figure 4, our experiments demonstrate that WE-SE consistently achieves a high speedup of at least 1.5x across all tested $\lambda$ values from 0.1 to 0.9. This indicates that in practical scenarios, regardless of what the optimal configuration of $\lambda$ is, WE-SE can achieve substantial acceleration.
> > >
> > > In CD setting, as discussed in Section 4.3 "Speedup ratio for different weight values of $\mu$ in CD" and illustrated in Figure 5, our experiments show that although the speedup of CD-SE gradually declines as increasing $\mu$ increases, CD-SE still offers a substantial acceleration. In addition, we report the performance of CD across a range of $\mu$ values in [CD across Different mu](https://anonymous.4open.science/r/SE-Rebuttal-Supplement-ICML25-667C/cd_across_different_mu.md). The results indicate that optimal performance is generally achieved at smaller $\mu$ values such as 0.1 or 0.2, which also deliver notable speedups.
> > >
> > > **2. The choice of $\lambda$ and $\mu$ in the main experimental setup**
> > >
> > > For our main results, we set $\lambda = 0.5$ and $\mu = 0.1$. This choice was guided not by a focus on speed, but by prior research and our empirical observations, aligning with values commonly used in practice.
> > >
> > > In WE setting, using average weights—such as $\lambda = 0.5$ for a two-model ensemble or $\lambda_i = 1/3$ for a three-model ensemble—is the most commonly adopted approach [1] [2] [3]. Huang et al. also noted in Section 3.5 of [2] that average weighting is "the most common practice".
> > >
> > > In CD setting, the results presented in [CD across Different mu](https://anonymous.4open.science/r/SE-Rebuttal-Supplement-ICML25-667C/cd_across_different_mu.md) show that CD consistently improves performance only when $\mu = 0.1$. Therefore, we adopt $\mu = 0.1$ in our main experiments to maintain consistency and facilitate comparison across results.
> > >
> > > **3. Clarification regarding Figure 8**
> > >
> > > As outlined in Section 4.1, "Ensemble Functions and Methods", and summarized in Table 1, our experiments mainly focus on two settings: *WE with models of comparable sizes* and *CD with models of different sizes*. Figure 8, however, presents results for *WE with models of different sizes*—a setting we view as a potential application of SE, rather than a central focus of this work.
> > >
> > > Notably, in the WE setting, as shown in Figure 4 and previously discussed, WE-SE achieves robust acceleration across all values of $\lambda$ without the need to be “carefully chosen”.
> > >
> > >
> > >
> > > [1] Breaking the Ceiling of the LLM Community by Treating Token Generation as a Classification for Ensembling
> > >
> > > [2] Ensemble Learning for Heterogeneous Large Language Models with Deep Parallel Collaboration
> > >
> > > [3] Determine-Then-Ensemble: Necessity of Top-k Union for Large Language Model Ensembling

---

### Official Review · Reviewer_A7ww · 2025-03-19

**Overall Recommendation:** 3

**Summary:**

This paper proposes Speculative Ensemble, accelerating ensemble speed without sacrificing the ensemble quality, inspired by speculative decoding. They theoretically prove the speed improvement over standard ensemble approaches. Experimental results also support their arguments and better ensemble efficiency.

**Claims And Evidence:**

While vanilla ensemble approaches require sequential calls of all models, the proposed methods smartly leverage Speculative decoding to make it more efficient. Theoretical guarantee for speedup is also supporting the claim very well.

**Essential References Not Discussed:**

N/A

**Experimental Designs Or Analyses:**

The validation is too focused on speed, and there's no performance trade-off from the ensemble. While it's clear that it's faster than existing ensembles, I'd like to see the performance gain from the ensemble as well. A clearer comparison of speed and performance with existing SD-based methodologies that used model ensembles would be helpful.

**Methods And Evaluation Criteria:**

- If the proposal and verifier differ a lot (i.e., small $c$), setting $\lambda$ larger than $\frac{c}{1+c}$ will naturally increase the acceptance rate and speed obviously. However, we know that there's a trade-off where performance might degrade as it converges towards the proposal model's performance. Using an ensemble, there could be an optimal $\lambda$ for peak performance, but this method doesn't seem to offer a way to find that optimal value. While ensemble intuitively speeds things up, uniform weights like 0.5 or 0.3 are likely near-optimal only when model sizes are similar. Combining small and large models might introduce inefficiency in finding the right trade-off.
- Regarding this, Figure 8 suggests that with differing model sizes, there isn't a clear optimal $\lambda$, and ensembling the proposal distributions doesn't seem to help performance. It's unclear if ensembling actually improved overall performance through this graph. In such settings (llama-3, -2, opt), a comparison of the accuracy-speedup trade-off with vanilla SD would be useful. Can you show a performance comparison between them given the same throughput?
- Also, if ensembling small model distributions is less effective, I don't see the benefits for verifying bonus tokens. Of course, it makes sense if it helps performance with similar sized models, and I think that efficient refinement of bonus tokens could be possible during autoregressive generation after the swap.

**Other Comments Or Suggestions:**

N/A

**Other Strengths And Weaknesses:**

N/A

**Questions For Authors:**

I’d be happy to discuss with the authors.

**Relation To Broader Scientific Literature:**

I found the concept of using SD for model ensembling interesting. However, I'm not sure how much gain there is in terms of speed and performance compared to existing methodologies in the SD literature that used combinations of two distributions (small-large).

**Theoretical Claims:**

I’ve reviewed the theoretical claims well. One question I have is regarding Corollary 3.5, which states that if the proposer and verifier are swapped (alternate proposal framework), there is acceleration when $\lambda < \frac{c}{1+c}$. Have these things been reflected upon or considered? It seems like the same $\lambda$ is used even when switching.

---

> ### Author Rebuttal · Authors · 2025-03-31
>
> We thank the reviewer for their time and insightful comments. Below, we address each concern in detail.
>
> For clarity and brevity, we use the following abbreviations: Methods and Evaluation Criteria (MAEC), Theoretical Claims (TC), Experimental Designs or Analyses (EDOA).
>
> **MAEC1: contrastive decoding is the ensemble function in our setup when model sizes differ**
>
> As discussed in Section 4.1 "Ensemble Functions and Methods" and summarized in Table 1, for model pairs like Llama-3, -2, and OPT, where the proposer and verifier differ a lot, we applied **contrastive decoding (CD)** [1] rather than the traditional weighted ensemble (WE).
>
> CD enhances token generation quality by subtracting the logits of a smaller model from those of a larger one, as defined in Equation (5). When combined with CD, LLMs generally achieve improved performance. To support this, we present results using Llama-3 and -2 with CD, and compare them to the large model baseline, as detailed in [CD](https://anonymous.4open.science/r/SE-Rebuttal-Supplement-ICML25-667C/cd.md).
>
> Therefore, when model sizes differ, our method is not intended to trade performance for speed. Instead, our focus is, when an ensemble method (e.g., CD) improves LLM performance, SE can further accelerate the ensemble while preserving these gains.
>
> Additionally, it is important to point out that, as shown in Section 4.1 ("Ensemble Functions and Methods") and Table 1, we also evaluated the speedup of SE under the WE setting when model sizes are similar. The results are presented in Table 2.
>
> **MAEC2: about weighted ensemble when model sizes differ in Figure 8**
>
> Although this paper primarily focuses on CD when model sizes differ, we also introduce WE with different model sizes as an exploratory approach to the quality-speed tradeoff. This discussion appears on page 4, line 203 (right column), with the corresponding results shown in Figure 8. However, this is not a central focus of the paper; rather, we present it as a potential strategy for achieving faster acceleration in vanilla SD—specifically, by applying WE with the proposal model and using SE.
>
> In this context, $\lambda$ serves as a hyperparameter that governs the quality-speed tradeoff. One possible selection strategy is to choose the largest $\lambda$ such that the ensemble performance exceeds a predefined threshold. As shown in Figure 8, a larger $\lambda$ indicates greater acceleration. Therefore, this selection strategy ensures that the highest speed is achieved while meeting the performance requirements.
>
> **MAEC3: concerns about verifying bonus tokens from small model**
>
> First, as previously noted, when model sizes differ, our primary focus is on CD. This ensemble approach can outperform a single large model. Therefore, verifying bonus tokens is meaningful.
>
> In the context of  WE with different model sizes, as you noted, verifying bonus tokens from the large model typically leads to reduced performance. Therefore, using the bonus tokens directly without verification is a more suitable approach. We appreciate your constructive feedback and will reflect it in the revised version. That said, our design prioritizes interpretability—specifically, ensuring that the distribution of generated tokens remains controllable and consistent. Consequently, in some cases, it remains beneficial to verify bonus tokens produced by the large model. For instance, as mentioned on page 5, line 231 (left column), some studies suggest that appropriately ensembling a smaller model can enhance safety.
>
> **TC: about Corollary 3.5**
>
> The term "swap" in Corollary 3.5 does not refer to the dynamic role alternation between proposer and verifier in the alternate proposal framework.
>
> Corollary 3.5 is specifically formulated for the Speculative-based Ensemble, which includes only the methods introduced in Section 3.2 and excludes the alternate proposal framework. It states that when the proposer and verifier are fixed, choosing an appropriate proposer model before inference begins can ensure acceleration. For example, when ensembling $\mathcal{M}_p$ and $\mathcal{M}_q$, if using $\mathcal{M}_p$ as the proposer does not guarantee acceleration, then $\mathcal{M}_q$ must. The "swap" referenced in Corollary 3.5 refers to this initial selection at the start of inference and does not occur dynamically during the inference process.
>
> Additionally, Corollary 3.7 guarantees the acceleration of SE under the alternate proposal framework. This corollary provides a more general result that ensures acceleration for any form of ensemble, not just WE.
>
> **EDOA & MAEC2: compare to single model SD**
>
> Regarding this, please refer to our response to Reviewer m8L7 under "Q1, 3, 4 & W3, 4".
>
>
>
> If you have any further questions or concerns, please feel free to let us know. We are committed to addressing any concerns to the best of our ability.
>
>
>
> [1] Contrastive Decoding: Open-ended Text Generation as Optimization

---

> > ### Comment · Reviewer_A7ww · 2025-04-04
> >
> > Thanks for the detailed explanation to address my questions. I missed the parts that CD is used when the model sizes differ.
> > All my concerns are resolved, and I believe this paper is above the acceptance bar.

---

> > > ### Author Response · Authors · 2025-04-05
> > >
> > > Thank you for the thoughtful and constructive feedback. We're glad that our responses addressed your concerns and appreciate your positive assessment. We will incorporate the rebuttal points into the revision to further improve the paper. If the clarifications merit a score revision, we would be grateful for your consideration.

---

### Official Review · Reviewer_hXqY · 2025-03-19

**Overall Recommendation:** 4

**Summary:**

This paper proposes "Speculative Ensemble", a method for speeding up auto-regressive generation from LLM ensembles using ideas from speculative decoding. For example, in the case of a two model ensemble, one can treat one of the models as the draft model, generate tokens with that model, and then process those tokens with the other model, with the important twist that during the verification algorithm the ensemble distribution is used as the target distribution (instead of directly using the "verifier's" distribution). This method can be extended to ensembles with > 2 models. Additionally, this method can be used to explore speed-quality trade-offs when using a weak model as the draft model and a strong model as the target model---as you assign larger weight to the draft model in the ensemble distribution, the acceptance rate and speed goes up, but the quality of the generation goes down.

Theoretically, the paper proves that speculative ensemble is always at least as fast as the naive ensembling approach of running each model independently.
Empirically, the paper shows across a variety of tasks that the proposed method attains 1.11x-2.23x speedup over the naive ensembling approach.

**Claims And Evidence:**

Yes, the claim are support by clear and convincing evidence across numerous model pairs and tasks.

**Essential References Not Discussed:**

I am not aware of any key references that weren't discussed.

**Experimental Designs Or Analyses:**

Yes, I did not see any issues.

**Methods And Evaluation Criteria:**

Yes, the proposed methods and evaluation criteria make sense. I found the "alternate proposal framework", as well as the extension of this method to N models, clever. The theoretical analysis is interesting and illuminates the speedups that should be expected from this method.

It would have been useful to report the "raw speeds" attained by the proposed method, to compare with SOTA in the literature.

**Other Comments Or Suggestions:**

It could be worth comparing with the very strong baseline of running the different models in the ensemble on difference devices, simply as a reference point (each model processes the last generated token in parallel, then the ensemble distribution is computed, and then a token is sampled from that distribution).

**Other Strengths And Weaknesses:**

The method is clever, novel, and well-validated.  I see no glaring weaknesses.

**Questions For Authors:**

In Tables 2 and 3, what exactly is the SD method? Does it simply throw away the bonus token?

**Relation To Broader Scientific Literature:**

To the best of my knowledge, this paper is the first to show how to use speculative decoding to sample from a distribution which is computed by combining two or more other model distributions. The method relates to the broader literature on speculative decoding and on how to speed up autoregressive generation from LLMs.

**Theoretical Claims:**

I (relatively quickly) checked the proofs of the claims in the main paper, and did not find any issues. The results make sense to me.

---

> ### Author Rebuttal · Authors · 2025-03-31
>
> We sincerely thank the reviewer for the time and effort spent reviewing our submission and greatly appreciate your insightful comments and constructive suggestions. Below, we have done our best to address each of your concerns in detail.
>
> **Methods And Evaluation Criteria: report the "raw speeds"**
>
> We sincerely thank you for your insightful suggestions. To better showcase SE's performance, we have reported raw generation speeds, as presented in [Raw Speed](https://anonymous.4open.science/r/SE-Rebuttal-Supplement-ICML25-667C/raw_speed.md).
>
> As you correctly noted, SE is the first method proposed to accelerate any form of LLM ensemble. We have extended SD baselines to the ensemble scenario where possible and included them in our comparisons. Other SOTA methods in SD, such as EAGLE [1] and Medusa [2], might also be adapted to ensemble scenario, but doing so would require specific modifications, which are beyond the scope of this work.
>
> **Other Comments Or Suggestions: compare with the very strong baseline**
>
> Thank you for your valuable suggestion. For clarity, we refer to the strong baseline you identified as the *parallel ensemble* (PE).
>
> First, we use the popular LLM ensemble repository yaoching0/GaC to test the speed of PE and implement a corresponding sequential ensemble to compute speedup. However, as shown in [Parallel Ensemble](https://anonymous.4open.science/r/SE-Rebuttal-Supplement-ICML25-667C/pe.md), PE is even slower than the sequential ensemble.
>
> The inefficiency likely stems from overly frequent communication. In particular, during the generation of each token, PE requires communication between the main node and two GPUs. Because the time to generate a single token is very short (especially when kv cache is enabled) the communication overhead becomes significant, introducing noticeable latency. In contrast, standard sequence-level parallelism avoids this issue, as the time to generate an entire sequence is much longer than the communication time, rendering the overhead negligible. These findings further underscore the importance of SE. Note that the speed reported in the original GaC repository was measured with kvcache disabled. In contrast, our tests were conducted with the kvcache enabled, as it is commonly used in current LLM inference. Therefore, our setup provides a more realistic evaluation.
>
> Additionally, the speedup achieved by PE is closely tied to its implementation and the underlying hardware. With improved engineering and more powerful hardware, PE could potentially attain greater speedup. Nonetheless, we maintain that SE remains a promising approach. For a detailed explanation, please see our response to Reviewer 3eWy under "Claims and Evidence 1".
>
> **Questions For Authors: the SD method in Tables 2 and 3**
>
> We apologize for the earlier lack of clarity.
>
> We use an example to clarify this SD process: In each cycle, suppose the proposal model $\mathcal{M}_q$ sequentially generates 5 tokens, which are then verified in parallel by the target model $\mathcal{M}_p$, resulting in 6 distributions, where the first 5 ones correspond to the verified distributions of 5 proposal tokens, while the 6-th is the bonus distribution. Since the 6-th bonus distribution lacks a corresponding distribution from $\mathcal{M}_q$ to compute the ensemble distribution, we cannot directly sample the bonus token from the ensemble distribution. To resolve this, we invoke $\mathcal{M}_q$ again to produce the 6-th distribution. Finally, we apply the standard accept-reject criterion in SD.
>
> Compared to this SD baseline, SE addresses this by treating the 6-th token as a proposal from $\mathcal{M}_p$ and verifies it using $\mathcal{M}_q$.  This verification not only obtains the 6-th distribution from $\mathcal{M}_q$, but also generates an additional bonus token from $\mathcal{M}_q$. In the next cycle, only 4 invocations to $\mathcal{M}_q$ are needed to form the proposal, thus reducing one invocation of $\mathcal{M}_q$ compared to the SD in Tables 2 and 3.
>
> As described above, the SD in Tables 2 and 3 does not throw away the bonus token; instead, it utilizes the token in the same manner as the standard SD. However, as previously discussed, SE utilizes bonus tokens more efficiently. With the same number of model invocations per cycle, it can generate one additional bonus token, resulting in greater acceleration.
>
>
>
> If you have any further questions or concerns, please feel free to let us know. We are committed to addressing any concerns to the best of our ability.
>
>
>
> [1] EAGLE-3: Scaling up Inference Acceleration of Large Language Models via Training-Time Test
>
> [2] Medusa: Simple LLM Inference Acceleration Framework with Multiple Decoding Heads

---

### Decision · Program_Chairs · 2025-05-01

**Decision:**

Accept (poster)

**Comment:**

Ensembling a set of LLMs has often shown to yield better quality than the individual LLMs in the pool. This paper proposes the use of speculative decoding (SD) to speed-up an ensemble of multiple models, with the smallest model used as the drafter and the ensembled distribution used to the verify the draft tokens. When comparing to sequential ensembles, the authors show notable speed-ups, while maintain quality neutrality.

The reviewers liked the application of SD to ensembles, but had concerns about the lack of comparisons to non-ensemble methods and questions about the relative sizes of the LLMs in the ensemble needing to be similar to provide quality gains. The authors provided additional experiments to partially address this concern.

While we are happy to accept the paper, we encourage the authors to include additional discussions contrasting their proposal with prior work:

**Contrasting with non-ensemble methods**: Based on the additional results, it appears that speculative ensembles (e.g. with weighted ensembling for similar sized models, and contrastive decoding for different sized models) offer ***better quality* but at a *lower speed-up* compared to non-ensemble methods** (e.g. vanilla SD, Medusa). We strongly urge the authors to include these additional results in the paper, and include an elaborate **discussion on comparison to non-ensemble methods and the role of model sizes**.

**Contrasting with similar prior work:** The authors need to differentiate their work from two closely related papers which came up during the post-rebuttal discussions:
- **Speculative Contrastive Decoding**, ACL 2024: The authors cite this paper but state that it differs in the use of "contrastive decoding for (only the) rejected tokens". I am not sure if this distinction is entirely accurate - my understanding is that this prior work also makes very similar changes to SD's acceptance criterion and residual distribution via contrastive decoding (lines 6 and 8 in their Alg. 1), but may differ in how it handles the bonus token. The main difference seems to be that the current paper handles more general ensembling functions.
- **Faster Cascades via Speculative Decoding**, ICLR 2025: This paper was not cited, and also makes a similar proposal as the present paper, except that the ensembling is done via a cascading rule instead of a simple weighting or contrastive decoding.

Both these papers need to be discussed more elaborately in the main text. In particular, it is important for the authors to differentiate their results with CD from the previous "Speculative CD" method, and consider including direct empirical comparisons to this method.